# 3D-PROPERTIES: IDENTIFYING CHALLENGES IN DPO AND CHARTING A PATH FORWARD

**Yuzi Yan**[1,3♮†] **Yibo Miao**[2,3♮†] **Jialian Li**[3]**, Yipin Zhang**[3]**, Jian Xie**[3]**, Zhijie Deng**[2*]**, Dong Yan**[3*]

[1]Department of Electronic Engineering, Tsinghua University
[2]Shanghai Jiao Tong University    [3]Baichuan AI
`yan-yz17@tsinghua.org.cn`, {`miaoyibo, zhijied`}`@sjtu.edu.cn`,
`lijialian7@163.com, zypzyp665@gmail.com, xiejian1990@gmail.com, sproblvem@gmail.com`

## ABSTRACT

Aligning large language models (LLMs) with human preferences has gained significant attention, with Proximal Policy Optimization (PPO) as a standard yet computationally expensive method and Direct Preference Optimization (DPO) as a more efficient alternative. While DPO offers simplicity, it remains underutilized in state-of-the-art LLMs, suggesting potential limitations. In this work, we revisit DPO, analyzing its theoretical foundations and empirical performance to bridge this gap. We identify three key properties—termed **3D**-properties—that emerge from DPO's learning process: **D**rastic drop in rejected response likelihood, **D**egradation into response suppression, and **D**ispersion effect on unseen responses. We show that these issues arise from DPO's optimization dynamics, where the interaction between chosen and rejected response gradients leads to instability. Our findings are supported by experiments on both a controlled toy model and real-world LLM tasks, including mathematical problem-solving and instruction following. To address these challenges, we propose simple regularization techniques that improve training stability and performance. Additionally, we examine how preference data distribution impacts DPO's effectiveness, offering insights into how alignment models handle out-of-domain (OOD) data. Our work connects these observations to broader research and provides a theoretical explanation for DPO's limitations. We hope these insights will guide future advancements in reward-model-free preference learning, bringing it closer to reward-model-based approaches.

## 1 INTRODUCTION

Large language models (LLMs) have demonstrated exceptional performance across a wide range of tasks and domains (Touvron et al., 2023; Chowdhery et al., 2023; Jiang et al., 2023; Zhang et al., 2022). Several techniques have been developed for fine-tuning LLMs, most notably Supervised Fine-Tuning (SFT) and Reinforcement Learning from Human Feedback (RLHF) (Achiam et al., 2023; Touvron et al., 2023). SFT involves directly training LLMs on labeled data to tailor their responses for specific tasks, whereas RLHF refines LLMs by incorporating feedback that aligns their outputs with human preferences. RLHF, in particular, has been instrumental in expanding the application of both closed-source (OpenAI, 2022; Anthropic, 2024; Team et al., 2023) and open-source LLMs (Touvron et al., 2023; Yang et al., 2023), driven by the need to align foundational models with human values and preferences (Ziegler et al., 2019; Stiennon et al., 2020; Ouyang et al., 2022).

Existing RLHF methods can be majorly categorized into two classes based on whether the reward signal is explicitly modeled. *Reward-model-based (RM-based) alignment* pioneered by OpenAI (Ouyang et al., 2022; Achiam et al., 2023; Touvron et al., 2023) first trains a Reward Model (RM) from user preferences, typically through Maximum Likelihood Estimation (MLE), and then

---

♮Equal contribution.
†This work was done during an internship at Baichuan AI.
*Corresponding authors.

leverages actor-critic algorithms such as Proximal Policy Optimization (PPO) (Schulman et al., 2017) to tune the SFT model to realize alignment. This approach often requires substantial computational resources and suffers from sample inefficiency (Choshen et al., 2019). Conversely, another class of methods, known as *reward-model-free (RM-free) alignment*, such as Direct Preference Optimization (DPO) (Rafailov et al., 2024), Identity Preference Optimization (IPO) (Azar et al., 2024), Sequence Likelihood Calibration (SLiC) (Zhao et al., 2023), DPO-positive (Pal et al., 2024) and Simple Preference Optimization (SimPO) (Meng et al., 2024), do not rely on an extra RM. These approaches offer a more resource-efficient alternative by optimizing the policy directly from preferences, therefore attracting much attention from the academic community, where computational resources are often limited.

In this work, we begin our analysis by using the vanilla DPO as a case study, subsequently extending our findings to encompass broader RM-free alignment strategies. Despite its simplicity and promise, DPO has exhibited several perplexing phenomena that remain unclear or underexplained in practice. One notable counter-intuitive observation is that the likelihood of both preferred and rejected responses tends to decrease over the course of DPO training (Yuan et al., 2024; Mitchell, 2023), while the likelihood of certain tokens diverging from the training data increases (Xu et al., 2024a). Additional observations are summarized in Section 2.1. Without a deeper theoretical exploration of these phenomena, purely empirical efforts to apply or improve DPO are likely to face inefficiencies.

Our work identifies the issues surrounding vanilla DPO and its variants from both theoretical and practical perspectives. The analysis reveals inherent instability in the DPO training process, which we encapsulate as the **3D**-properties: **D**rastic drop in the likelihood of rejected responses, **D**egradation into response suppression, and **D**ispersion effect on unseen responses. Through our analytical framework, we show that these phenomena stem from the inherent features of DPO's optimization objective, where the interaction between the gradients of chosen and rejected responses leads to instability and hinders overall performance. Furthermore, our findings confirm that the distribution of preference data critically influences DPO's effectiveness, with on-policy DPO performing better than off-policy DPO, which is consistent with concurrent empirical studies (Tang et al., 2024; Guo et al., 2024).

To enhance DPO's stability and performance, we propose several regularization methods, including the adaptive adjustment of weights on the gradients of chosen and rejected responses, as well as incorporating an SFT loss into the objective. Our results suggest a fundamental trade-off within the DPO algorithm: balancing the mitigation of the 3D-properties while preventing LLMs from straying too far from the preference learning paradigm. Additionally, we compare DPO with the state-of-the-art RM-based method, RLHF-PPO, revealing that its superiority stem largely from avoiding the 3D-properties. Our experimental approach begins with the design of a toy model to quickly validate our hypotheses, followed by a rigorous test of the actual performance of real LLMs on tasks such as mathematical problem solving and instruction following.

As this topic has garnered significant attention recently, an increasing number of works are contributing to the discussion. To highlight the contributions of our approach, we compare our findings with several of the most relevant concurrent studies in Section 2.2. A comprehensive review of related works is provided in Appendix A.

## 2 PRELIMINARIES

**Large Language Model (LLM).** An LLM defines a $\theta$-parameterized conditional distribution $\pi_\theta(a|x)$, which takes a prompt $x$ as input and produces a response $a$. More specifically, the sampling from LLMs is performed in an auto-regressive manner, $\pi_\theta(a|x) = \prod_t \pi_\theta(a_t|x, a_{1:t-1})$, where $a_t$ is the $t$-th token in the response $a$ and $a_{1:t-1}$ are tokens in the response before $a_t$.

**RM-based RLHF.** Training LLMs typically involves three stages: Pretraining, SFT, and RLHF. We outline the standard PPO paradigm here, which is a typical RM-based RLHF algorithm. Beginning with a well-trained SFT model, denoted as $\pi_0$, we proceed by sampling two responses from $\pi_0$ for each instance in a given prompt set. Subsequently, we compile a preferece dataset $\mathcal{D} = \{(x, a^+, a^-)\}$, where $a^+$ and $a^-$ denote human-preferred and human-dispreferred completions, respectively. The distribution of the preference dataset is assumed to follow the Bradley-Terry

model (Bradley & Terry, 1952), i.e., the probability of response $a^+$ is better than $a^-$ is given by:

$$p_r(a^+ \succ a^-|x) = \frac{\exp(r(x, a^+))}{\exp(r(x, a^+)) + \exp(r(x, a^-))} = \sigma(r(x, a^+) - r(x, a^-)), \quad (1)$$

where $\succ$ represents the preference relation, and $\sigma(x) = \frac{1}{1+e^{-x}}$ is the sigmoid function. To train a RM $r(\cdot, \cdot)$, we maximize the log-likelihood of the observed preferences by minimizing the following loss function:

$$\ell_R(r) = - \sum_{(x, a^+, a^-)} \log p_r(a^+ \succ a^-|x) = - \sum_{(x, a^+, a^-)} \log \sigma(r(x, a^+) - r(x, a^-)). \quad (2)$$

During the reinforcement learning phase, we update the LLM to maximize the return from the learned RM using the following objective function:

$$\max_\theta J_r(\theta) = \max_\theta \sum_x \mathbb{E}_{a \sim \pi_\theta(\cdot|x)} \left[ r(x, a) - \beta \log \frac{\pi_\theta(a|x)}{\pi_0(a|x)} \right], \quad (3)$$

where $\pi_\theta$ is initialized as $\pi_0$ and $\beta$ controls the deviation from the original model. PPO (Schulman et al., 2017) is typically used to solve the problem in practice. Algorithms that optimize the policy using a separate RM are referred to as *RM-based* alignment.

**DPO.** Instead of learning a separate RM, DPO (Rafailov et al., 2024) directly optimizes the policy $\pi_\theta$ over preference data. DPO implicitly leverages a particular choice of RM parameterization that enables the extraction of its optimal policy in closed form, without a reinforcement learning training loop:

$$\ell^{\text{DPO}}(\theta) = - \sum_{(x, a^+, a^-)} \log \sigma \left[ \beta \log \frac{\pi_\theta(a^+|x)}{\pi_0(a^+|x)} - \beta \log \frac{\pi_\theta(a^-|x)}{\pi_0(a^-|x)} \right]. \quad (4)$$

As shown, DPO leverages logistic regression loss to directly fine-tune the LLM on preference data. This approach, along with its various variants (Zhao et al., 2023; Amini et al., 2024; Azar et al., 2024), is referred to as *RM-free* alignment due to the elimination of an explicit RM.

## 2.1 UNDEREXPLORED OBSERVATIONS ABOUT DPO

Though the absence of the need for additional RM training makes DPO particularly attractive, several observations remain underexplored. The most concerning issue is, to the best of our knowledge, few models using DPO (or other RM-free algorithm) have achieved performance comparable to the state-of-the-art closed-source LLMs such as OpenAI's GPT-4o or Anthropic's Claude, which reportedly use PPO methods during training. Besides, many other phenomena have been reported but lack comprehensive theoretical explanations. Here we make a summary for clarity.

**Observation 1.** During the vanilla DPO training, the likelihood of both the chosen and rejected responses in the preference datasets tends to decrease, whereas the likelihood of unseen tokens not appearing in the preference pairs tends to increase (Mitchell, 2023).

**Observation 2.** Compared with RM-based alignment, the performance of DPO is relatively unstable and sub-optimal (Wang et al., 2024).

**Observation 3.** The performance of DPO is significantly affected by the distribution shift between the model outputs and the preference dataset. In general, on-policy DPO, where both the chosen responses and the rejected responses are sampled from the policy model $\pi_\theta$, outperforms other scenarios (Tang et al., 2024).

## 2.2 COMPARISON WITH RELATED CONTEMPORARY STUDIES

Several concurrent studies attempt to explain these observations, and we highlight our differences to underscore our contributions. For Observation 1, Feng et al. (2024) shares similar points of view regarding the degradation on gradients but offers limited analysis and lacks experimental validation on real LLMs. In contrast, our work offers a more rigorous and comprehensive analysis supported by theoretical insights and validation across toy models and large-scale real-world LLM experiments.

For Observation 2, Xu et al. (2024a) point out that the policy minimizing the PPO loss is a subset of that minimizing the DPO loss, which offers a partial explanation. However, their analysis focuses solely on the endpoint of the optimization and does not examine the dynamic process by which the policy evolves. It leaves unaddressed how unexpected policies emerge during training. In contrast, our gradient analysis offers a comprehensive understanding of the entire optimization trajectory, shedding light on how and why sub-optimal policies arise throughout the DPO training process. For Observation 3, Tang et al. (2024) investigate the performance gap between on-policy and off-policy alignment algorithms from an empirical perspective, while our insights are rooted in theoretical findings.

Our work advances the understanding of these observations, providing critical insights into the underlying mechanisms and reinforcing the findings of these concurrent studies. In the following section, we will present our theoretical explanations for these observations.

## 3 FUNDAMENTAL LIMITATIONS OF VANILLA DPO: 3D-PROPERTIES

We first identify a critical flaw inherent in vanilla DPO. At first glance, the loss function of vanilla DPO, as defined in Eq. (4), appears to be composed of two parts: the term $\log \frac{\pi_\theta(a^+|x)}{\pi_0(a^+|x)}$ aims to increase the likelihood of chosen responses, while the term $\log \frac{\pi_\theta(a^-|x)}{\pi_0(a^-|x)}$ seeks to decrease the likelihood of rejected responses. However, this seemingly straightforward interpretation overlooks significant underlying issues, which we characterize through the 3D-properties of vanilla DPO.

**Property 1** (**D**rastic drop in rejected response likelihood)**.** The likelihood of a rejected response tends to shift much more rapidly than that of a chosen response.

**Property 2** (**D**egradation into response suppression)**.** As optimization progresses, DPO gradually loses its ability to steer the direction of optimizing chosen responses and instead devolves into merely suppressing the rejected responses.

**Property 3** (**D**ispersion effect on unseen responses)**.** As DPO training progresses, the likelihood of both chosen and rejected responses gradually decreases, while the likelihood of generating out-of-distribution (OOD) responses increases.

These properties are non-trivial, as they reveal inherent challenges in DPO's optimization process that are not immediately apparent from the loss function alone. These phenomena closely align with the empirical observations we discussed earlier, pointing to the structural limitations of DPO. In the following section, we delve into a theoretical analysis to further explain the origins of these 3D-properties and provide insights into their impact on the optimization trajectory.

### 3.1 THEORETICAL FOUNDATIONS

In this sections, we provide the theoretical foundations for the 3D-properties, followed by detailed explanations of the observations discussed in Section 2.1. The loss function for DPO as shown in Eq. (4) can be re-written by:

$$\ell^{\text{DPO}}(\theta) = \sum_{(x,a^+,a^-)} \log\left(1 + \left(\frac{\pi_0(a^+|x)}{\pi_0(a^-|x)} \cdot \frac{\pi_\theta(a^-|x)}{\pi_\theta(a^+|x)}\right)^\beta\right). \tag{5}$$

For a given triple $(x, a^+, a^-)$, let

$$\alpha := \left(\frac{\pi_0(a^+|x)}{\pi_0(a^-|x)}\right)^\beta, \quad \pi^+ := \pi_\theta(a^+|x), \quad \pi^- := \pi_\theta(a^-|x), \quad z := \frac{\pi_\theta(a^-|x)}{\pi_\theta(a^+|x)} = \frac{\pi^-}{\pi^+}.$$

Then we have

$$\frac{\partial \ell^{\text{DPO}}}{\partial \pi^+} = \frac{\partial \log(1 + \alpha z^\beta)}{\partial z} \frac{\partial z}{\partial \pi^+} = \frac{\alpha\beta}{1 + \alpha z^\beta} z^{\beta-1} \frac{\partial z}{\partial \pi^+} = \frac{\alpha\beta}{1 + \alpha z^\beta} z^{\beta-1} \left[-\frac{\pi^-}{(\pi^+)^2}\right],$$

$$\frac{\partial \ell^{\text{DPO}}}{\partial \pi^-} = \frac{\partial \log(1 + \alpha z^\beta)}{\partial z} \frac{\partial z}{\partial \pi^-} = \frac{\alpha\beta}{1 + \alpha z^\beta} z^{\beta-1} \frac{\partial z}{\partial \pi^-} = \frac{\alpha\beta}{1 + \alpha z^\beta} z^{\beta-1} \left(\frac{1}{\pi^+}\right),$$

with these simplified forms, we can obtain the following corollaries to explain the properties above:

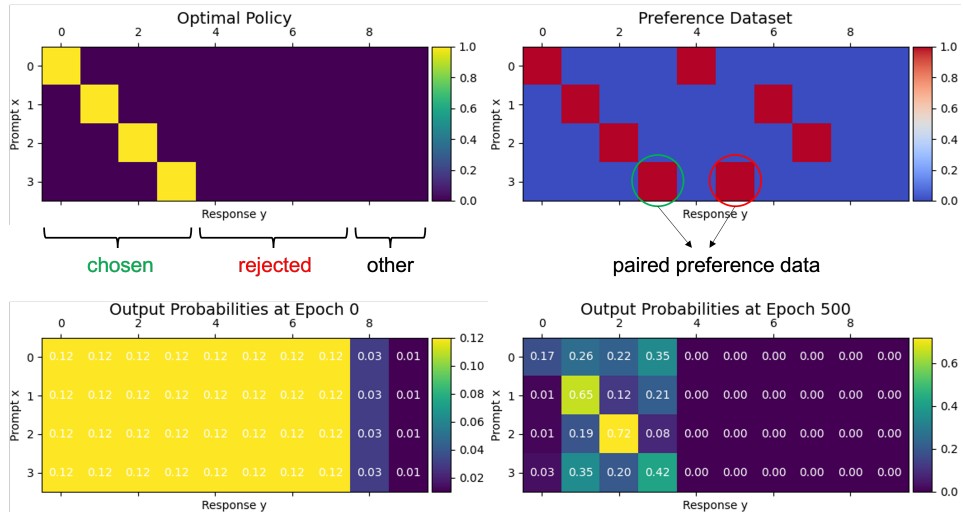

Figure 1: Toy model setup. Top left: the optimal policy where the highlighted blocks represent optimal responses. Top right: preference dataset construction. Lower left: the initialization of the SFT model. Lower right: policy output after DPO training.

**Corollary 1** (Explanation for Property 1). The ratio of the gradient with respect to the rejected response likelihood $\pi^-$ to the gradient with respect to the chosen response likelihood $\pi^+$ is equal to the ratio of $\pi^+$ to $\pi^-$:

$$\frac{\partial \ell^{\text{DPO}}}{\partial \pi^-} / \frac{\partial \ell^{\text{DPO}}}{\partial \pi^+} = \frac{\partial z}{\partial \pi^-} / \frac{\partial z}{\partial \pi^+} = -\frac{\pi^+}{\pi^-},$$

which indicates that as $\pi^+$ increases and $\pi^-$ decreases, the gradient with respect to $\pi^-$ grows faster, leading to a more rapid decline in the likelihood of the rejected response.

**Corollary 2** (Explanation for Property 2). As $\pi^- \to 0$, we have $z \to 0$ and $\frac{\alpha\beta}{1+\alpha z^\beta} \to \alpha\beta$. Thus,

$$\frac{\partial \ell^{DPO}}{\partial \pi^+} \to -\alpha\beta(\pi^+)^{-\beta-1}(\pi^-)^\beta \to 0, \quad \frac{\partial \ell^{DPO}}{\partial \pi^-} \to \alpha\beta(\pi^+)^{-\beta}(\pi^-)^{\beta-1} \to \infty,$$

given that $\beta < 1$ and $\pi^- \to 0$. This creates a dynamic where the gradient for the rejected response grows exceedingly large, while the gradient for the chosen response diminishes significantly. As a result, DPO progressively shifts its focus to suppressing the rejected responses and loses the ability to steer the direction of optimizing chosen responses.

**Corollary 3** (Explanation for Property 3). When $\pi^-$ drastically drops to 0, the gradient on $\pi^+$ fails and the likelihood of the chosen response is likely to decrease along with the rejected response as they often share many similar tokens and patterns. The constancy of the sum of probabilities implies that as both $\pi^+$ and $\pi^-$ decrease, the likelihood will randomly disperse into other unseen responses out of the preference dataset.

Based on these theoretical insights, we can further explore and explain the observations discussed in Section 2.1. Observation 1 is directly explained by Corollary 3. For Observation 2, we will show that 3D-properties do not manifest during the RM training process in the RM-based alignment pipeline. Regarding Observation 3, we will demonstrate that the distribution gap between the LLM's original outputs and the preference dataset plays a crucial role in determining the influence of the 3D-properties. The impact of these properties is notably less pronounced in on-policy DPO, where the preference dataset is sampled directly from the policy model's outputs. In the following sections, we will delve deeper into each of these statements and provide empirical validation.

## 3.2 SYNTHETIC VALIDATION WITH A TOY MODEL

In this section, we introduce a simplified toy model specifically designed to facilitate synthetic experiments, thereby enhancing the persuasiveness of our arguments from Corollary 1 to 3. Then we conduct experiments on real LLMs.

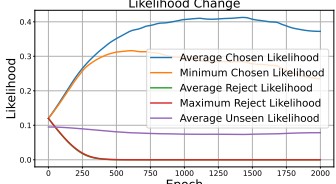 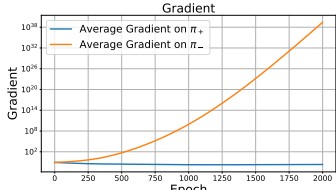 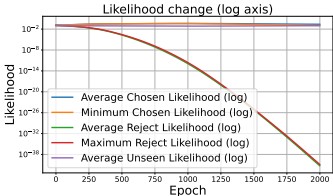

Figure 2: Dynamic optimization process with vanilla DPO using the toy model. Left: likelihood dynamics over training epochs. The blue curve represents the average likelihood of chosen responses, yellow shows the minimum for chosen responses, green represents the average for rejected responses, red shows the maximum for rejected responses, and purple represents the average for unseen responses. Middle: dynamics of averaged $\frac{\partial \ell^{DPO}}{\partial \pi^+}$ and $\frac{\partial \ell^{DPO}}{\partial \pi^-}$ over training epochs. Right: likelihood dynamics over training epochs on a log scale, highlighting the drastic drop in the likelihood of rejected responses.

### 3.2.1 TOY MODEL SETUP

The diagram for the toy model is illustrated in Figure 1. We construct a discrete space consisting of 4 prompts and 10 responses. The policy $\pi_\theta$, which simulates a simplified version of an LLM, is implemented as a three-layer MLP that processes a one-hot vector and outputs a categorical distribution over the responses. The response space is organized such that the first 4 dimensions correspond to chosen responses, dimensions 5 through 8 represent rejected responses, and the final 2 dimensions correspond to unseen responses not present in the preference dataset.

In this setup, each prompt has an optimal response (e.g., response 1 is optimal for prompt 1, as shown in the upper left figure). When constructing the preference dataset for DPO training, we adopt a mini-batch sampling strategy to mimic real-world annotation processes. Specifically, assuming an ideal annotator, each input prompt is perfectly matched with its optimal response—corresponding to the diagonal elements of the matrix, as illustrated in the upper right figure in Figure 1. For each mini-batch, we then randomly select one other response within the batch to create preference data pairs. This approach ensures that gradient updates are computed from diverse mini-batch samples.

To simulate the Pretraining and SFT process, we manually assign output probabilities and use them as labels to train $\pi_\theta$. Initially, as shown in the lower left figure, we set the likelihood of both chosen and rejected responses at $0.12$, treating both as on-policy. The constructed preference dataset is then used for DPO training, with the output after 500 epochs shown in the lower right. The code is provided in the supplementary material.

### 3.2.2 RESULTS

Figure 2 illustrates the dynamic optimization process during DPO training. In the first figure, the likelihood of chosen responses (blue and yellow curves) increases, while the likelihood of rejected responses (green and red curves) decreases in the early phases of training. However, as training progresses, the likelihood of chosen responses begins to decline in the longer run. During this degradation phase, as both chosen and rejected response likelihoods decrease, the probability is redistributed to unseen responses (purple curve).

The second and third figures reveal the underlying causes of this shift: as $\pi_\theta(a^-|x)$ approaches zero, the absolute value of $\partial \ell^{\mathrm{DPO}}/\partial \pi^-$ increases sharply compared to $\partial \ell^{\mathrm{DPO}}/\partial \pi^+$. The absolute value of $\partial \ell^{\mathrm{DPO}}/\partial \pi^+$ becomes progressively smaller, weakening its influence on the optimization direction. These results align with the earlier theoretical analysis.

Moreover, this insight provides an explanation for the observed superiority of on-policy DPO (Observation 3). In contrast to off-policy DPO, on-policy DPO begins with a higher likelihood for rejected responses, thereby extending the duration before their likelihood significantly diminishes. To further validate the differing impacts of on-policy and off-policy DPO, we configure four scenarios by adjusting the initial distribution of outputs to simulate these conditions. A higher initialized likelihood (0.12) simulates responses sampled in an on-policy manner, while a lower one (0.02) simulates responses sampled off-policy. The initial state and the subsequent changes in the likelihood

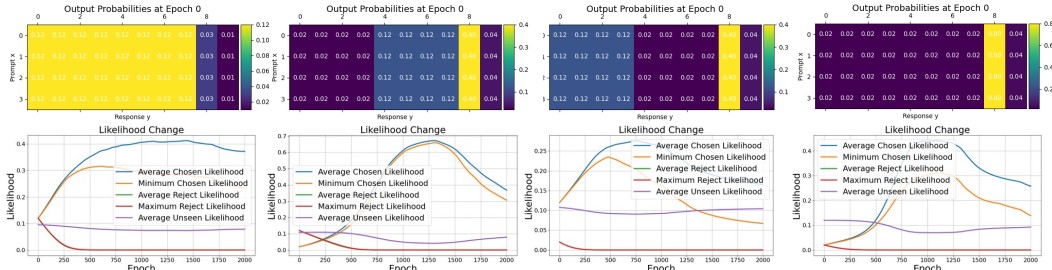

Figure 3: From left to right, the figures show the initial state and the likelihood dynamics for chosen/rejected/unseen responses in Scenarios 1 to 4, similar to the left diagram in Figure 2: (1) both chosen and rejected responses are on-policy, (2) chosen off-policy and rejected on-policy, (3) chosen on-policy and rejected off-policy, and (4) both off-policy.

of each response are illustrated in Figure 3. Notably, in Scenario 1, where both chosen and rejected responses are on-policy, the 3D-properties are relatively mild, as shown by the high peak probability of the optimal response (approximately 0.6) and the minimal dispersion effect on unseen responses. Additional detailed results and analyses for all four scenarios can be found in Appendix D.

**The intention of the toy model and its connection to real LLMs.** The toy model serves as an abstract simulation that amplifies the effect of 3D-properties, which are less pronounced and harder to visualize in real-world experiments. While the toy model differs from real LLM training in several ways—such as sampling frequency—its design offers useful insights. In real-world settings, DPO is typically trained over one epoch, with each data point used only a few times. In contrast, in the toy model, the same data points are sampled repeatedly. Conceptually, this is similar to treating each input/output as a token rather than a complete prompt/response, where each token may be sampled multiple times during real-world training. Since both the chosen and rejected responses are generated from the same prompt, they often share common tokens. As a result, the decrease in the likelihood of rejected responses can impact the likelihood of chosen responses, leading to a corresponding decline in their likelihood.

### 3.3 REGULARIZATION TECHNIQUES

It becomes evident that the rate at which $\pi_\theta(a^-|x)$ declines is crucial in determining the severity of the 3D-properties' impact. This observation leads to the following proposition:

**Proposition 1.** To lessen the severity of the 3D-properties, it is advantageous to moderate the rate at which the likelihood of rejected responses declines.

Inspired by Proposition 1, we introduce two straightforward regularization techniques. The first technique employs adaptive values of $\beta$ to control the rate at which the likelihood of rejected responses declines, referred as Flex-DPO. The second technique involves augmenting the DPO loss with an SFT loss, a strategy that has been shown to significantly enhance the stability of DPO in previous studies (Hou et al., 2024; Xu et al., 2024b). These regularization methods have shown promising results with our toy model and will be further validated in real LLMs in the following section. The theoretical analysis is similar to that of vanilla DPO thus deferred to Appendix B.2.

### 3.4 INHERENT ABSENCE OF 3D-PROPERTIES IN RM-BASED ALIGNMENT

In this section, we show that 3D-properties do not manifest in RM-based alignment methods, which may account for why DPO methods only achieve sub-optimal performance. Since DPO is closely related to RM training, and the Best-of-N performance of the RM can partially reflect the ultimate performance of the policy model (Gui et al., 2024), we focus on analyzing the RM's objective. For a given $(x, a^+, a^-)$, let $r^+ := r(a^+|x)$ and $r^- := r(a^-|x)$, the gradients with respect to $r^+$ and $r^-$ are:

$$\frac{\partial \ell^{\mathrm{RM}}}{\partial r^+} = \frac{\partial \log(1 + e^{(r^- - r^+)})}{\partial r^+} = -\frac{e^{(r^- - r^+)}}{1 + e^{(r^- - r^+)}} = -\frac{1}{1 + e^{(r^+ - r^-)}},$$

$$\frac{\partial \ell^{\mathrm{RM}}}{\partial r^-} = \frac{\partial \log(1 + e^{(r^- - r^+)})}{\partial r^-} = \frac{e^{(r^- - r^+)}}{1 + e^{(r^- - r^+)}} = \frac{1}{1 + e^{(r^+ - r^-)}}.$$

This indicates that the gradients for the chosen and rejected responses are balanced and do not exhibit 3D-properties. In Section 4.5, we will further discuss the relationship between DPO and RM-based alignment in real LLMs.

# 4 EXPERIMENTS

In this section, we transition from theoretical analyses and toy model simulations to real-world experiments with LLMs to further validate our theoretical insights. We verify the existence of 3D-properties, the superiority of on-policy DPO over off-policy DPO, the superiority of RM over DPO, and the effectiveness of the proposed regularization technique.

## 4.1 EXPERIMENTAL SETUP

**Datasets.** We chose mathematical reasoning and instruction following as our primary benchmarks because these tasks are easily quantifiable, providing clear metrics for evaluating model performance. For mathematical reasoning, we used MATH (Hendrycks et al., 2021) as the main dataset for both training and testing[1]. To assess the model's out-of-distribution (OOD) generalization capabilities, we selected SuperCLUE-Math (Xu et al., 2020), another dataset which was used exclusively for testing. Additionally, we included two in-house datasets focused on poem and slogan generation to evaluate the model's ability to handle creative tasks with strict structural and linguistic constraints. The poem dataset, for instance, which has rigid format and rhyme requirements, making it a good test for evaluating the model's ability to follow complex instructions. Further details and descriptions of the datasets used are provided in Appendix C.

It is widely accepted in industry that preference datasets for model alignment should cover a broad range of domains. Following this consensus, we further utilized a general dataset consisting of approximately 400,000 preference samples across diverse domains. These prompts were sourced from HH-rlhf (Bai et al., 2022) and UltraFeedBack (Cui et al., 2024). A detailed breakdown of the dataset sizes is provided in Table 6 in Appendix C.

**The LLMs of concern.** We focus on Baichuan2-13B and Baichuan2-33B, an advanced bilingual (Chinese and English) LLM series. The 13B model is openly available (Yang et al., 2023), and the 33B model extends the 13B architecture with increased parameters.

## 4.2 EFFECT OF TRAINING DATA DISTRIBUTION: ON-POLICY VS. OFF-POLICY

Building on the theoretical insights in Section 3, we hypothesize that the performance of vanilla DPO is significantly influenced by the distribution gap between the training dataset and outputs of the policy model, specifically whether the algorithm is on-policy or off-policy. Off-policy DPO uses an external preference dataset, while on-policy DPO samples preferences directly from the policy model. On-policy DPO enjoys a smaller distribution gap compared with off-poliy DPO.

To conduct on-policy DPO, we used the policy model to produce 8 candidates for each prompt in the train set of MATH. The best and worst responses were selected by GPT-4 (Achiam et al., 2023) to form a preference pair, with the standard solutions given as the reference context. After filtering out uniformly good or bad responses, we compiled the MATH* dataset, which contains 5,826 pairs $\{x, a^+, a^-\}$. We randomly selected 2,000 samples from the original test set to serve as the test set for MATH*. For off-policy DPO, we used the original solutions from the dataset as the chosen responses, and generated the rejected responses using Qwen1.5-7B (Bai et al., 2023), a relatively earlier model with limited capabilities.

To validate the hypothesis that the presence of off-policy data weakens performance, we implemented the four scenarios consistent with the toy model (Figure 3). We evaluated the policy model using GPT-4, which assigned scores ranging from 1 to 5 based on the accuracy of both the final answer and the problem-solving process, with also the standard solutions given as the reference

---

[1]Only the prompts from the dataset were used to generate the preference dataset; further details are provided in Section 4.2.

Table 1: Results tested on MATH* and SuperCLUE-Math. Scenario 1, with both chosen and rejected responses sampled on-policy, shows the best performance.

| Setting | Baichuan2-13B | | Baichuan2-33B | |
|---|---|---|---|---|
| | 5 points | 4&5 points | 5 points | 4&5 points |
| basemodel | 32.237% | 42.539 % | 44.485% | 53.229% |
| DPO in Scenario 1 | **37.132%** | **47.082%** | **47.465%** | **54.759%** |
| DPO in Scenario 2 | 32.860% | 43.445% | 44.216% | 51.409% |
| DPO in Scenario 3 | 28.323% | 41.576% | 44.473% | 53.924% |
| DPO in Scenario 4 | 26.833% | 37.685% | 46.618% | 54.648% |

Table 2: Test results on the self-built Poem and Slogan datasets. All metrics are evaluated such that higher values indicate better performance.

| | Poem | | | | | Slogan | |
|---|---|---|---|---|---|---|---|
| | Row Number | Words per Row | Rhythm | Tone Pattern | Title | Word Count | Content |
| Base | 0.75 | 0.61 | 0.64 | 0.60 | 0.51 | 0.34 | 0.57 |
| PPO | 0.91 | **0.79** | **0.87** | **0.82** | **1** | **0.47** | **0.78** |
| DPO | **0.93** | 0.75 | 0.83 | 0.75 | 0.78 | 0.45 | 0.70 |

context. The scoring criteria are detailed in Table 4. The average performance on the MATH* and SuperCLUE-Math datasets is reported in Table 1, with specific results in Table 8. Among the four scenarios, Scenario 1—where both chosen and rejected responses are on-policy—ensured a more stable DPO training process and delivered the best performance.

Additionally, we report the log probabilities before and after training in Table 7 in Appendix D.2. According to Proposition 1, the key factor affecting the impact of 3D-properties is the decline rate of rejected responses' likelihood, $\log \pi(a^-)$. Scenario 1 shows the slowest decline in likelihood compared to the other scenarios, effectively mitigating the adverse effects of 3D-properties, which explains the superior performance of on-policy DPO in our tests.

We also plot the gradients during the DPO training process for Scenario 1, as shown in Figure 6 in the Appendix D.2. This visualization supports the analysis in Section 3, demonstrating that the gradients for rejected responses increase more rapidly during training. This excessive decline in the likelihood of generating rejected responses can ultimately lead to model degradation.

In addition to pure on-policy and off-policy DPO, Scenario 2, where the chosen response is off-policy and the rejected response is on-policy, is also prevalent in industry. For instance, in math problems, researchers often treat the correct dataset solution as the chosen response and the LLM-generated incorrect answer as the rejected one, which we demonstrate to be detrimental. Scenario 3 is a mirror experiment for Scenario 2. These experiments confirm that incorporating off-policy data into the preference training set degrades DPO performance.

### 4.3 EXPERIMENTAL VALIDATION OF REGULARIZATION TECHNIQUES

Following Flex-DPO, the regularization methods outlined in Section 3.3, we fixed $\beta^+$ and systematically decreased $\beta^-$. As indicated by the gradient analysis in Appendix B.2.1, the gradient of rejected responses with respect to $\beta^-$ follows a non-monotonic trajectory, initially increasing and then decreasing. Reducing $\beta^-$ on the left side of this extreme point can effectively reduce the gradient magnitude. However, indiscriminately minimizing the gradient is not always advantageous. As illustrated in Figure 4, model performance does not consistently improve with an excessively small $\beta^-$ (see the trend with $\beta^- < 0.08$). Over-reduction of $\beta^-$ risks causing the DPO algorithm to deviate from the preference learning

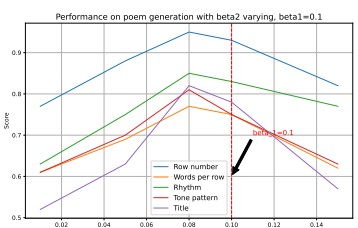

Figure 4: Performance on poem generation, $\beta^-$ varying with $\beta^+ = 0.1$.

paradigm and regress toward behavior akin to the SFT algorithm, ultimately compromising its generalization capabilities. This finding warrants further investigation, and while preliminary insights are discussed in Appendix D, deeper exploration is needed. This aspect will be addressed in future research.

Additionally, we tested other DPO variants, such as IPO and SLiC, on the MATH* and SuperCLUE-Math datasets, with results presented in Table 9. Flex-DPO consistently outperforms vanilla DPO, IPO and SLiC, highlighting the effectiveness of the proposed regularization techniques.

### 4.4 RELATIVE INSTABILITY OF DPO TRAINING COMPARED TO RM TRAINING

To assess the stability and performance gap between DPO and RM training, we conducted a parallel comparison using identical datasets. Both models were built on the Baichuan2-33B architecture and trained on the HH-rlhf and UltraFeedback datasets, with evaluations conducted on the HH-rlhf and MATH datasets. The primary evaluation metric was accuracy, defined as the proportion of instances where the model correctly identified the chosen response as superior to the rejected one.

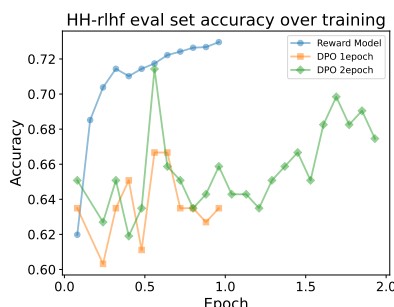

Figure 5: Accuracy of RM and DPO on HH-rlhf eval set over the training process.

As shown in Figure 5, RM training proved to be significantly more stable, whereas DPO training exhibited notable fluctuations. These findings are consistent with the theoretical results in Section 3.4, which indicate that 3D-properties are absent in RM-based alignment methods. Furthermore, as illustrated in Figure 8 and Figure 9 in Appendix D.2, the DPO model demonstrated a higher tendency to overfit. Specifically, the sharp deceleration in accuracy improvement after the second epoch suggests that the model was overfitting the training data, highlighting the more aggressive optimization dynamics of DPO.

### 4.5 SUBOPTIMALITY OF DPO COMPARED TO RM-BASED ALIGNMENT

To further compare the performance of DPO and the end-to-end RM-based alignment (PPO), we tested both approaches on two datasets for poem and slogan generation. These datasets serve as ideal benchmarks for evaluating instruction-following capabilities, given their explicit and structured scoring criteria. For poem creation, the model must generate responses in accordance with specific text and tone formats based on the prompt. Evaluation metrics include five key aspects: *Row Number*, *Words per Row*, *Rhythm*, *Tone Pattern*, and *Title*. For slogan creation, evaluation is based on *Word Count* and *Content*. Using Baichuan2-33B for our experiments, the results, shown in Table 2, demonstrate that DPO underperforms compared to RLHF-PPO on both datasets.

## 5 CONCLUSIONS

In this study, we conducted a comprehensive theoretical analysis to elucidate why DPO does not perform as well as RM-based alignment algorithm. The principal challenge identified in DPO is summarized as 3D-properties. We substantiated our theoretical framework through experimental results obtained from both a toy model and real LLMs in practical applications, including mathematical reasoning and instruction following. Additionally, we assessed the effectiveness of specific regularization techniques. Furthermore, by contrasting DPO training with RM training, we highlighted the inherent instability of DPO. We hope this work could offer research directions to narrow the gap between RM-free preference learning methods and RM-based ones. We leave the discussion of limiation to Appendix D.3.

### ACKNOWLEDGEMENTS

This work was supported by NSF of China (Nos. 92470118, 62306176) and Natural Science Foundation of Shanghai (No. 23ZR1428700).

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

## A  DETAILED BACKGROUND AND RELATED WORKS

Large language models (LLMs) are profoundly transforming the way we work and live. Performing a three-stage process is the default practice for training LLMs: Pretraining, Supervised Fine-Tuning (SFT), and Reinforcement Learning from Human Feedback (RLHF). The roles of Pretraining and SFT are broadly understood: Pretraining encodes knowledge and SFT aligns question-answer formats. Relatively speaking, the understanding of RLHF is relatively insufficient. Specifically, Direct Preference Optimization (DPO) and its variants, as reward-model-free algorithms, have garnered significant attention due to their elegant mathematical form and relatively low resource requirements (Rafailov et al., 2024; Pal et al., 2024; Guo et al., 2024; Xiong et al., 2023). However, it has also sparked considerable debate because of its unstable performance in practical applications (Li et al., 2023; Xu et al., 2024a).

### A.1  ON-POLICY ALIGNMENT VS. OFF-POLICY ALIGNMENT

The key inspiration for the DPO algorithm (Rafailov et al., 2024) is a closed-form solution to the RL step in RLHF, and thus an equivalent solution to the optimal policy for RLHF objective. The original DPO work is an off-policy learning algorithm for it relies on an extra preference dataset (Helpful-and-Harmless (Bai et al., 2022)), where the preference pairs are not generated by the policy LLM itself. On the other hand, there are a bunch of on-policy learning algorithms developed, where the preference responses are sampled from the policy model. Guo et al. (2024) proposed the on-policy version of DPO. In on-policy DPO, all responses are sampled in a batch-wise way. A natural trade-off between them is the iterative DPO introduced by Xiong et al. (2023); Xu et al. (2024a). The algorithm begins by initializing with an additional preference dataset, then iteratively trains a policy using DPO, collects response pairs through exploration policies, obtains preference signals from human or AI labelers, and updates the dataset with the newly labeled data.

## A.2 Insights into DPO

Though the RM-free algorithms are favored due to their lower computational overhead, if they can achieve on-performance with state-of-art RM-based methods such as RLHF-PPO sparked a lot of discussions. Liu et al. (2023) proves that the absence of RM in DPO constrains its ability to sample preference pairs from the optimal policy. Xu et al. (2024a) show that DPO may have fundamental limitations that its optimal solution is a superset of the optimal solution of the PPO algorithm. This work also reports the empirical results that the performance of DPO is affected by the distribution shift between the model outputs and the preference dataset. Feng et al. (2024) discusses the limitations of DPO from the perspective of gradient numerical stability, and conducted experiments to preliminarily verify it. However, they did not conduct experiments on real LLM and illustrate the correlation.

## A.3 Other RM-free alignment algorithms

A major limitation of the DPO objective is its reliance on the Bradley-Terry model to convert pairwise preferences into point-wise rewards. To overcome this, Azar et al. (2024) introduced $\Phi$-preference optimization ($\Phi$PO), where DPO is a special case of it that $\Phi(P) = \log \frac{P}{1-P}$. Identity-preference optimization (IPO) is a variant that replaces the $\Phi$-function by an identity mapping function $\Phi(P) = P$.

Different from the DPO or IPO, the core idea of Sequence Likelihood Calibration (SLiC) (Zhao et al., 2023) is to calibrate the likelihood of ranked sequences sampled from the policy being trained. The SLiC loss function can be decomposed into two parts: the rank function to guarantee that the difference between $\log \pi_\theta(a^+|x)$ and $\log \pi_\theta(a^-|x)$ is greater than $\delta$ under the current policy $\pi_\theta$, and the cross-entropy regularizer that to encourage the model to stay close to the SFT model.

There are some other variants that tries to improve DPO, such as KTO (Ethayarajh et al., 2024), NCA (Chen et al., 2024), ODPO (DPO with an offset) (Amini et al., 2024). KTO uses a Kahneman-Tversky model of human utility and proposes a method that directly maximizes the utility of generations instead of maximizing the log-likelihood of preferences. NCA leverages Noise Contrastive Estimation (NCE) to bridge the gap in handling reward datasets explicitly annotated with scalar evaluations. ODPO does not treat every preference pair equally during fine-tuning and requires the difference between the likelihood of the preferred and dispreferred response to be greater than an offset value.

## B Theoretical Foundations

### B.1 Fundamental Limitation in Vanilla DPO

Here we revisit the theoretical findings in Section 3.1 in detail. The loss function for vanilla DPO is given by

$$\ell^{\text{DPO}}(\theta) = \sum_{(x,a^+,a^-)} \log \left( 1 + \left( \frac{\pi_0(a^+|x)}{\pi_0(a^-|x)} \frac{\pi_\theta(a^-|x)}{\pi_\theta(a^+|x)} \right)^\beta \right).$$

For a given $(x, a^+, a^-)$, let

$$\alpha := \left( \frac{\pi_0(a^+|x)}{\pi_0(a^-|x)} \right)^\beta, \quad \pi^+ := \pi_\theta(a^+|x), \quad \pi^- := \pi_\theta(a^-|x), \quad z := \frac{\pi_\theta(a^-|x)}{\pi_\theta(a^+|x)}.$$

Then we have

$$\frac{\partial \ell^{\text{DPO}}}{\partial \pi^+} = \frac{\partial \log(1 + \alpha z^\beta)}{\partial z} \frac{\partial z}{\partial \pi^+} = \frac{\alpha\beta}{1 + \alpha z^\beta} z^{\beta-1} \frac{\partial z}{\partial \pi^+},$$

$$\frac{\partial \ell^{\text{DPO}}}{\partial \pi^-} = \frac{\partial \log(1 + \alpha z^\beta)}{\partial z} \frac{\partial z}{\partial \pi^-} = \frac{\alpha\beta}{1 + \alpha z^\beta} z^{\beta-1} \frac{\partial z}{\partial \pi^-}.$$

Considering the case when $\pi^- \to 0$, we get $(\alpha\beta)/(1 + \alpha z^\beta) \to \alpha\beta$, thus,

$$\frac{\partial \ell^{\text{DPO}}}{\partial \pi^+} \to -\alpha\beta(\pi^+)^{-\beta-1}(\pi^-)^\beta, \quad \frac{\partial \ell^{\text{DPO}}}{\partial \pi^-} \to \alpha\beta(\pi^+)^{-\beta}(\pi^-)^{\beta-1}.$$

As $\pi^- \to 0$, since $\beta < 1$, $\frac{\partial \ell^{\text{DPO}}}{\partial \pi^+}$ is proportional to $(\pi^-)^\beta$ and tends to 0, while $\frac{\partial \ell^{\text{DPO}}}{\partial \pi^-}$ is proportional to $(\pi^-)^{\beta-1}$ and tends to infinity. Therefore, in this case, the gradient for the rejected action becomes extremely large, while the gradient for the chosen action becomes very small.

Then we want to further explore the token-level gradient. Here, $\pi^+$ and $\pi^-$ are the likelihood of the sequences. Let $\pi_i^+$ be the current selection probability of the $i$-th token for the chosen sequence, and let $\pi_i^-$ be the current selection probability of the $i$-th token for the rejected sequence:

$$\pi^+ = \prod_i \pi_i^+ = \pi_{-i}^+ \cdot \pi_i^+, \quad \pi^- = \prod_i \pi_i^- = \pi_{-i}^- \cdot \pi_i^-,$$

where we have $\frac{\partial \pi}{\partial \pi_i} = \pi_{-i}$. Consider a softmax function, $s_i = \frac{e^{z_i}}{\sum_j e^{z_j}}$, the corresponding gradients are

$$\frac{\partial s_i}{\partial z_i} = s_i(1 - s_i), \quad \frac{\partial s_j}{\partial z_i} = -s_i s_j, \quad i \neq j.$$

Let

$$C(\pi^+, \pi^-) := \alpha \beta^+ (\pi^+)^{-\beta^+} (\pi^-)^{\beta^-},$$

then, consider the current selection probability $\pi_i^+$ of the $i$-th token for the chosen sequence, let the sampled token's index be $c$. The logit corresponding to this token $c$ is denoted as $x_{i,c}^+$, then we have

$$\frac{\partial \ell^{\text{DPO'}}}{\partial x_{i,c}^+} = \frac{\partial \ell^{\text{DPO'}}}{\partial \pi^+} \frac{\partial \pi^+}{\partial \pi_i^+} \frac{\partial \pi_i^+}{\partial x_{i,c}^+} \to -C(\pi^+, \pi^-)(1 - x_{i,c}^+),$$

if $c' \neq c$, we have

$$\frac{\partial \ell^{\text{DPO'}}}{\partial x_{i,c'}^+} = \frac{\partial \ell^{\text{DPO'}}}{\partial \pi^+} \frac{\partial \pi^+}{\partial \pi_i^+} \frac{\partial \pi_i^+}{\partial x_{i,c'}^+} \to C(\pi^+, \pi^-)x_{i,c'}^+.$$

Similarly, consider the current selection probability $\pi_i^-$ of the $i$-th token for the rejected sequence, let the sampled token's index be $c$. The logit corresponding to this token $c$ is denoted as $x_{i,c}^-$, then we have

$$\frac{\partial \ell^{\text{DPO'}}}{\partial x_{i,c}^-} = \frac{\partial \ell^{\text{DPO'}}}{\partial \pi^-} \frac{\partial \pi^-}{\partial \pi_i^-} \frac{\partial \pi_i^-}{\partial x_{i,c}^-} \to C(\pi^+, \pi^-)(1 - x_{i,c}^-),$$

if $c' \neq c$, we have

$$\frac{\partial \ell^{\text{DPO'}}}{\partial x_{i,c'}^-} = \frac{\partial \ell^{\text{DPO'}}}{\partial \pi^-} \frac{\partial \pi^-}{\partial \pi_i^-} \frac{\partial \pi_i^-}{\partial x_{i,c'}^-} \to -C(\pi^+, \pi^-)x_{i,c'}^-.$$

We can see that the token-level gradients from the chosen response and the rejected response are at the same scale level. This reflects that DPO may not cause gradient numerical instability in the generation of a single token. However, if the impact of the algorithm on the state transition probability generated by autoregression is comprehensively considered, 3D-properties will still affect the performance of the algorithm.

## B.2 ANALYSIS ON REGULARIZATION TECHNIQUES

In Section 3.3, we propose two straightforward regularization techniques. Here we provide theoretical analysis to see why they can mitigate 3D-properties.

### B.2.1 FLEXIBLE $\beta$-DPO

The first technique employs variable values of $\beta$ to control the rate at which the likelihood of rejected responses declines. Consider using different $\beta^+$ and $\beta^-$ for the chosen and rejected responses:

$$\ell^{\text{flex-DPO}}(\theta) = - \sum_{(x,a^+,a^-)} \log \sigma \left[ \beta^+ \log \frac{\pi_\theta(a^+|x)}{\pi_0(a^+|x)} - \beta^- \log \frac{\pi_\theta(a^-|x)}{\pi_0(a^-|x)} \right].$$

The loss function can be re-written by:

$$\ell^{\text{flex-DPO}}(\theta) = \sum_{(x,a^+,a^-)} \log \left( 1 + \left( \frac{\pi_0(a^+|x)}{\pi_\theta(a^+|x)} \right)^{\beta^+} \left( \frac{\pi_\theta(a^-|x)}{\pi_0(a^-|x)} \right)^{\beta^-} \right).$$

For a given $(x, a^+, a^-)$, let

$$\alpha := \frac{\pi_0(a^+|x)^{\beta^+}}{\pi_0(a^-|x)^{\beta^-}}, \quad \pi^+ := \pi(a^+|x), \quad \pi^- := \pi(a^-|x), \quad z := \frac{\pi_\theta(a^-|x)^{\beta^-}}{\pi_\theta(a^+|x)^{\beta^+}}.$$

Then we have

$$\frac{\partial \ell^{\text{DPO'}}}{\partial \pi^+} = \frac{\partial \log(1 + \alpha z)}{\partial z} \frac{\partial z}{\partial \pi^+} = \frac{\alpha}{1 + \alpha z} \frac{\partial z}{\partial \pi^+},$$

$$\frac{\partial \ell^{\text{DPO'}}}{\partial \pi^-} = \frac{\partial \log(1 + \alpha z)}{\partial z} \frac{\partial z}{\partial \pi^-} = \frac{\alpha}{1 + \alpha z} \frac{\partial z}{\partial \pi^-}.$$

Considering the case when $\pi^- \to 0$, we get $\alpha/(1 + \alpha z) \to \alpha$, thus

$$\frac{\partial \ell^{\text{DPO'}}}{\partial \pi^+} \to -\alpha\beta^+ (\pi^+)^{-\beta^+ - 1} (\pi^-)^{\beta^-}, \quad \frac{\partial \ell^{\text{DPO'}}}{\partial \pi^-} \to \alpha\beta^- (\pi^+)^{-\beta^+} (\pi^-)^{\beta^- - 1}.$$

As shown by the expressions for the gradients of the DPO' objective with respect to the chosen and rejected response likelihoods, the magnitude of the gradients is controlled by the parameters $\beta^+$ and $\beta^-$. In particular, increasing $\beta^+$ strengthens the gradient for the chosen response, $\pi^+$, while reducing $\beta^-$ dampens the gradient for the rejected response, $\pi^-$, as it approaches zero. This gradient behavior suggests that adjusting these parameters can effectively mitigate the 3D-properties discussed in Section 3. Specifically, a large $\beta^+$ ensures that the likelihood of the chosen responses remains sufficiently reinforced, while a small $\beta^-$ prevents the likelihood of rejected responses from decreasing too rapidly, which would otherwise lead to the instability and degradation described earlier.

By fine-tuning $\beta^+$ and $\beta^-$, it becomes possible to control the interaction between the gradients of the chosen and rejected responses, reducing the drastic drop in rejected response likelihood, the degradation into response suppression, and the dispersion effect on unseen responses. This strategy thus offers a potential solution for improving the stability and performance of DPO by reducing the severity of the 3D-properties.

### B.2.2  SFT Loss Regularization

The second technique involves augmenting the DPO loss with an SFT loss, a strategy that has been shown to significantly enhance the stability of DPO in previous studies. We can rewrite the loss function:

$$\ell^{\text{SFT-DPO}}(\theta) = -\sum_{(x, a^+, a^-)} \left\{ \log \sigma \left[ \beta \log \frac{\pi_\theta(a^+|x)}{\pi_0(a^+|x)} - \beta \log \frac{\pi_\theta(a^-|x)}{\pi_0(a^-|x)} \right] - \gamma \log \pi_\theta(a^+|x) \right\} \quad (6)$$

Similarly, we have,

$$\frac{\partial \ell^{\text{SFT-DPO}}}{\partial \pi^+} = \frac{\partial \log(1 + \alpha z^\beta)}{\partial z} \frac{\partial z}{\partial \pi^+} = \frac{\alpha\beta}{1 + \alpha z^\beta} z^{\beta-1} \frac{\partial z}{\partial \pi^+} - \gamma \frac{1}{\pi^+},$$

$$\frac{\partial \ell^{\text{SFT-DPO}}}{\partial \pi^-} = \frac{\partial \log(1 + \alpha z^\beta)}{\partial z} \frac{\partial z}{\partial \pi^-} = \frac{\alpha\beta}{1 + \alpha z^\beta} z^{\beta-1} \frac{\partial z}{\partial \pi^-}.$$

As $\pi^- \to 0$, the gradient for the chosen action $\frac{\partial \ell^{\text{SFT-DPO}}}{\partial \pi^+} \to -\gamma \frac{1}{\pi^+} \neq 0$, meaning that the likelihood of the chosen responses can continue to be optimized. This behavior is significant, as it indicates that even as the likelihood of rejected responses $\pi^-$ approaches zero, the chosen response $\pi^+$ can still be improved. This is a key advantage of SFT-DPO over the vanilla DPO, which suffers from gradient vanishing for $\pi^+$ when $\pi^- \to 0$. The negative impact of 3D-properties is thus reduced, allowing for more stable and effective optimization in the long run.

### B.3  Other invariants of DPO

### B.3.1  Identity-preference Optimization (IPO)

In IPO, the loss function can be written by,

$$\ell^{\text{IPO}}(\theta) = \sum_{(x, a^+, a^-)} \left[ \log \left[ \frac{\pi_\theta(a^+|x)\pi_0(a^-|x)}{\pi_\theta(a^-|x)\pi_0(a^+|x)} \right] - \frac{1}{2\eta} \right]^2 \quad (7)$$

We directly give the gradients:

$$\frac{\partial \ell^{\text{IPO}}}{\partial \pi_\theta(a^-|x)} = -2 \left[ \log \left[ \frac{\pi_\theta(a^+|x)\pi_0(a^-|x)}{\pi_\theta(a^-|x)\pi_0(a^+|x)} \right] - \frac{1}{2\eta} \right] \cdot \frac{1}{\pi_\theta(a^-|x)} \tag{8}$$

$$\frac{\partial \ell^{\text{IPO}}}{\partial \pi_\theta(a^+|x)} = 2 \left[ \log \left[ \frac{\pi_\theta(a^+|x)\pi_0(a^-|x)}{\pi_\theta(a^-|x)\pi_0(a^+|x)} \right] - \frac{1}{2\eta} \right] \cdot \frac{1}{\pi_\theta(a^+|x)} \tag{9}$$

### B.3.2 SEQUENCE LIKELIHOOD CALIBRATION (SLiC)

In SLiC, the loss function can be written by,

$$\ell^{\text{SLiC}}(\theta) = \sum_{x,a^+,a^-} \max \left[ 0, \delta - \log \pi_\theta(a^+|x) + \log \pi_\theta(a^-|x) \right] - \eta \cdot \log \pi_\theta(a^+|x) \tag{10}$$

if $\delta > \log \frac{\pi_\theta(a^+|x)}{\pi_\theta(a^-|x)}$:

$$\frac{\partial \ell^{\text{SLiC}}}{\partial \pi_\theta(a^+|x)} = -\frac{1+\eta}{\pi_\theta(a^+|x)}, \quad \frac{\partial \ell^{\text{SLiC}}}{\partial \pi_\theta(a^-|x)} = \frac{1}{\pi_\theta(a^-|x)} \tag{11}$$

else:

$$\frac{\partial \ell^{\text{SLiC}}}{\partial \pi_\theta(a^+|x)} = -\frac{\eta}{\pi_\theta(a^+|x)}, \quad \frac{\partial \ell^{\text{SLiC}}}{\partial \pi_\theta(a^-|x)} = 0 \tag{12}$$

### B.3.3 SIMPLE PREFERENCE OPTIMIZATION (SIMPO)

In SimPO, the loss function is written by,

$$\ell^{\text{SimPO}}(\theta) = -\sum_{x,a^+,a^-} \left[ \log \sigma \left( \frac{\beta}{|a^+|} \log \pi_\theta(a^+|x) - \frac{\beta}{|a^-|} \log \pi_\theta(a^-|x) - \gamma \right) \right], \tag{13}$$

where $|\cdot|$ represents the length of the generated response and $\gamma$ is a target reward margin term. Similarly, we directly give the gradients:

$$\frac{\partial \ell^{\text{SimPO}}}{\partial \pi_\theta(a^+|x)} = -\frac{\beta}{|a^+| \cdot \pi_\theta(a^+|x)} \sigma \left( -\left( \frac{\beta}{|a^+|} \log \pi_\theta(a^+|x) - \frac{\beta}{|a^-|} \log \pi_\theta(a^-|x) - \gamma \right) \right), \tag{14}$$

$$\frac{\partial \ell^{\text{SimPO}}}{\partial \pi_\theta(a^-|x)} = \frac{\beta}{|a^-| \cdot \pi_\theta(a^-|x)} \sigma \left( -\left( \frac{\beta}{|a^+|} \log \pi_\theta(a^+|x) - \frac{\beta}{|a^-|} \log \pi_\theta(a^-|x) - \gamma \right) \right). \tag{15}$$

The ratio of the gradient is

$$\frac{\partial \ell^{\text{SimPO}}}{\partial \pi^-} \Big/ \frac{\partial \ell^{\text{SimPO}}}{\partial \pi^+} = -\frac{|a^+| \cdot \pi_\theta(a^+|x)}{|a^-| \cdot \pi_\theta(a^-|x)}, \tag{16}$$

which indicates that as $\pi^+$ increases and $\pi^-$ decreases, the gradient with respect to $\pi^-$ grows faster and leads to a rapid drop in the likelihood of the rejected response. However, different from DPO, as $\pi^- \to 0$, we have $\partial \ell^{\text{SimPO}}/\partial \pi^+ \to 0$. As for $\partial \ell^{\text{SimPO}}/\partial \pi^-$, the analysis is non-trivial. We conclude it as a lemma.

**Lemma 1.** As $\pi_\theta(a^-|x) \to 0$, the limit of the partial derivative regarding $\pi_\theta(a^-|x)$ in SimPO is:

$$\lim_{\pi_\theta(a^-|x) \to 0} \frac{\partial \ell^{\text{SimPO}}}{\partial \pi_\theta(a^-|x)} = \begin{cases} 0, & \text{if } \beta > |a^-|, \\ +\infty, & \text{if } \beta < |a^-|, \\ \frac{\beta C}{|a^-|}, & \text{if } \beta = |a^-|, \end{cases} \tag{17}$$

where $C = e^{-\frac{\beta}{|a^+|} \log \pi_\theta(a^+|x) + \gamma}$ is a constant independent of $\pi_\theta(a^-|x)$.

Table 3: Data source of the responses in 4 scenarios.

|  | chosen responses source | rejected responses source |
|---|---|---|
| Scenario 1 | Baichuan2-33B | Baichuan2-33B |
| Scenario 2 | Solutions from dataset | Baichuan2-33B |
| Scenario 3 | Baichuan2-33B | Qwen-7B |
| Scenario 4 | Solutions from dataset | Qwen-7B |

*Proof.* Let $\pi^- = \pi_\theta(a^-|x)$ and $\pi^+ = \pi_\theta(a^+|x)$. The partial derivative becomes:

$$f(\pi^-) = \frac{\beta}{|a^-| \cdot \pi^-} \cdot \sigma(-z),$$

where $z = \frac{\beta}{|a^+|} \log \pi^+ - \frac{\beta}{|a^-|} \log \pi^- - \gamma$.

As $\pi^- \to 0$, $\log \pi^- \to -\infty$, thus $z \to +\infty$. The sigmoid function approximates to:

$$\sigma(-z) \approx e^{-z} = C \cdot (\pi^-)^{\frac{\beta}{|a^-|}},$$

where $C = e^{-\frac{\beta}{|a^+|} \log \pi^+ + \gamma}$. By substituting back, we have:

$$f(\pi^-) \approx \frac{\beta C}{|a^-|} \cdot (\pi^-)^{\frac{\beta}{|a^-|}-1}.$$

Taking the limit as $\pi^- \to 0$:

$$\lim_{\pi^- \to 0} f(\pi^-) = \begin{cases} 0, & \text{if } \frac{\beta}{|a^-|} - 1 > 0 \ (\beta > |a^-|), \\ +\infty, & \text{if } \frac{\beta}{|a^-|} - 1 < 0 \ (\beta < |a^-|), \\ \frac{\beta C}{|a^-|}, & \text{if } \frac{\beta}{|a^-|} - 1 = 0 \ (\beta = |a^-|). \end{cases}$$

$\square$

Note that in practice, $\beta$ is normally chosen less than $|a^-|$, which means the drastic drop in rejected response will happen and 3D-properties still occur.

**Remark 1.** These variants all alleviate the 3D-properties problem of DPO, so the results on mathematical reasoning are partly improved (see Table 9). The invariants we tested do not perform well uniformly on all the tasks. For example, on instruction following tasks like poem generation, vanilla DPO outperforms SLiC and IPO. One hypothesis is their solution forms diverge significantly from the Bradley-Terry model, leading to a loss of generalization in preference learning.

## C  DATASET DESCRIPTION

Table 3 shows the data source for the LLM experiments in Section 4 in 4 scenarios. For example, in Scenario 1, the chosen and the rejected responses are both sampled from Baichuan2-33B and can be regarded as on-policy learning. In Scenario 4, the chosen responses are exactly the solutions given in the datasets while the rejected responses are sampled from another different LLM: Qwen-7B. In the experiments regarding Baichuan2-13B, we use the same data rather than re-sample the on-policy chosen responses. There are two reasons: 1. The 13B model is not as strong as the 33B model, therefore we can not sample enough high-quality responses as the chosen ones. 2. Models in the Baichuan2-series are all using the same dataset in Pretraining and SFT, therefore we can approximately think that their outputs are identically distributed. The log probability for the base model in Table 7 confirms this fact.

Table 4 describes the evaluation criteria of the responses to the math questions. A score of 5 means both the process and the result are correct, and a score of 4 or 5 means the answer is correct. We use these two indicators to evaluate the mathematical reasoning ability of the model. GPT-4 is used as the AI evaluator. We provide the evaluation prompt in our code in the supplementary material.

Table 4: Evaluation criteria of the responses to the math questions.

| | |
|---|---|
| 5 points | Full-score answer, requiring a correct response with the correct process, considering all possibilities, and being comprehensive. |
| 4 points | For complex questions, the answer is correct but lacks the process; for simple questions, the answer is correct but accompanied by a very redundant and verbose reasoning process. |
| 3 points | The answer is incorrect, but most of the process is correct, or the answer is correct, but there are obvious errors in the process. |
| 2 points | The answer is incorrect, and most of the process is incorrect. |
| 1 point | The answer and the entire process and thought process are incorrect, or the answer doesn't process to final result. |

Table 5: Poem dataset test set.

| Poem type | Quatrain | Song Ci | Ancient Poetry | Metrical poetry | Modern poetry |
|---|---|---|---|---|---|
| Test set Count | 138 | 518 | 93 | 173 | 85 |

Table 5 shows the types of different poems in the poem dataset we used and their corresponding numbers in the test set. The language is all Chinese. Chinese poetry has strict format and rhyme requirements depending on the type. For example, for the quatrains, the row number must be 4, the number of words per row must be 5 or 7. The second and fourth sentences in the quatrain are required to rhyme, that is, the words at the end of the second and fourth sentences need to follow the prescribed tone. In this manner, we design a rule-based evaluation system to score each dimension of the generated answers. We selected the following characteristics as the basis for our evaluation:

- *Row Number*: For quatrain, the row number must be 4. For metrical poetry, the row number must be 8.
- *Words per Row*: For quatrain and metrical poetry, the number of words per row must be 8.
- *Rhythm*: Every type of poetry has a certain rhyme pattern requirement. Since it is a bit complicated to describe case by case, we put the requirements in the form of code in the supplementary material.
- *Tone Pattern*: For Song Ci, the tone pattern depends on the brand name.
- *Title*: Determined by the requirement in the prompt.

For the Slogan dataset, we evaluate the model's performance based on whether it meets the word count requirements (*Word Count*) and the quality of the content (*Content*). We also provide the scoring and evaluation rule-based criteria in our code in the supplementary material.

Table 6 shows the amount of data in each dataset. MATH, SuperCLUE, UltraFeedBack and HH-rlfh are open-source datasets, while Poem, Slogan are in-house self-built datasets. We provide part of the in-house datasets in the supplementary material to clarify the format and the content. SuperCLUE is only for cross-dataset testing, and COMMON is only used for the comparison between the RM training and DPO training in Section 4.4.

# D EXPERIMENTS DETAILS

## D.1 TRAINING SETTING

In this section, we provide a detailed overview of our training settings. Following the implementation of Rafailov et al. (2024), we use the Adam optimizer with the learning rate set to 5e-7. The most sensitive parameter in the DPO algorithm is $\beta$ (and learning rate but less significant). Here we use the default setting aligned with the original DPO paper, The $\beta$ here is set to be 0.1 and the learning

Table 6: The statistic of used datasets.

|  | MATH* | SuperCLUE | Poem | Slogan | UltraFeedBack | HH-RLHF |
|---|---|---|---|---|---|---|
| train set | 5,826 | - | 93,269 | 13,592 | 170,000 | 336,820 |
| test set | 2,000 | 1,072 | 1,000 | 1,000 | - | - |

rate is set to be $5 \times 10^{-6}$, which is the best hyperparameter set as far as we explored. We set the batch size to 80 and the number of gradient accumulation steps to 2. The training epoch was set to 1. In IPO training, we set $\eta$ to be 0.1. In SLiC training, we set $\delta = 5$, $\eta = 0.1$. All experiments were conducted on a cluster consisting of 40 A100 GPUs.

### D.2 SUPPLEMENTARY EXPERIMENTAL RESULTS

Table 7 reports the log probabilities before and after training. Scenario 1 exhibits the slowest decline in likelihood compared to other scenarios, effectively mitigating the adverse effects of 3D-properties.

Table 8 show the performance enhancement in MATH and SuperCLUE by vanilla DPO training respectively. It is easy to see that Scenario 1 where both the chosen responses and the rejected responses are on-policy performs best.

Figure 4 and Table 9 represent the additional results on DPO variants and regularization techniques. It can be seen that DPO variants can achieve on-par or better performance compared with vanilla DPO. Figure 4 shows that as $\beta^-$ decreases, the performance in poem generation initially improves, reaches the peak point at around $\beta^- = 0.08$, and subsequently declines. The initial improvement is intuitive. We conjecture that an excessively low $\beta^-$ may cause DPO to perform like SFT on the chosen responses, thereby reducing its generalization capabilities. For different tasks, the peak point of $\beta^-$ can be different. For example, in mathematical reasoning, we can set $\beta^-$ to be $0.01$ and achieve better performance than vanilla DPO.

Figure 6 shows the change of the absolute value of gradient for the chosen and rejected responses ($\partial \ell^{\mathrm{DPO}}/\partial \pi^+$ and $\partial \ell^{\mathrm{DPO}}/\partial \pi^-$) during the training process of DPO on MATH. It can be seen that $|\partial \ell^{\mathrm{DPO}}/\partial \pi^-| \gg |\partial \ell^{\mathrm{DPO}}/\partial \pi^-|$, and the increasing rate of $\partial \ell^{\mathrm{DPO}}/\partial \pi^-$ is much higher than that of $\partial \ell^{\mathrm{DPO}}/\partial \pi^+$, which is align with Property 1.

Figure 8 and Figure 9 show the DPO convergence process with model trained on MATH and HH-rlhf respectively. In the second epoch, the accuracy growth of the model slows down sharply, which indicates that the model overfits the training data. The results further confirm that DPO is an aggressive optimization strategy compared to RLHF-PPO, and makes the impact of 3D-properties more prominent.

Among the four scenarios, Scenario 1, where both chosen and rejected responses are on-policy, ensures a more stable DPO training process and delivers the best testing performance, as shown in Table 1. As shown in Table 7, Scenario 1 exhibits the slowest decline in likelihood compared to other scenarios, effectively mitigating the adverse effects of 3D-properties. This explains the superior performance of on-policy DPO in our tests.

In addition to pure on-policy and off-policy DPO, Scenario 2 where the chosen response is off-policy and the rejected response is on-policy is also commonly seen in the industry. For example, in some math problems, researchers used to treat the accurate solution in the dataset to be the chosen response and the wrong answer generated by the LLM itself to be the rejected response, which we show is harmful. Scenario 3 is a mirror experiment to Scenario 2. These additional experimental results confirm that as long as the off-policy data is mixed into the training preference dataset, the performance of DPO will be weakened.

### D.3 LIMITATIONS

Despite the insights provided in this study, several limitations remain. First, while our theoretical analysis and experiments highlight the 3D-properties as a key factor in DPO's suboptimal performance, the complexity of real-world LLMs may involve additional factors that were not fully explored. Second, our experimental evaluation, though spanning diverse tasks such as mathematical

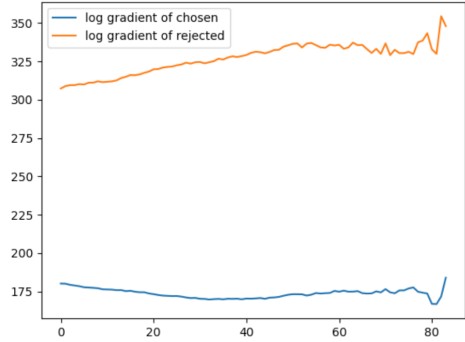

Figure 6: When training with on-policy data, the absolute value of the gradient for rejected responses increases, while the absolute value of the gradient for chosen responses remains almost unchanged.

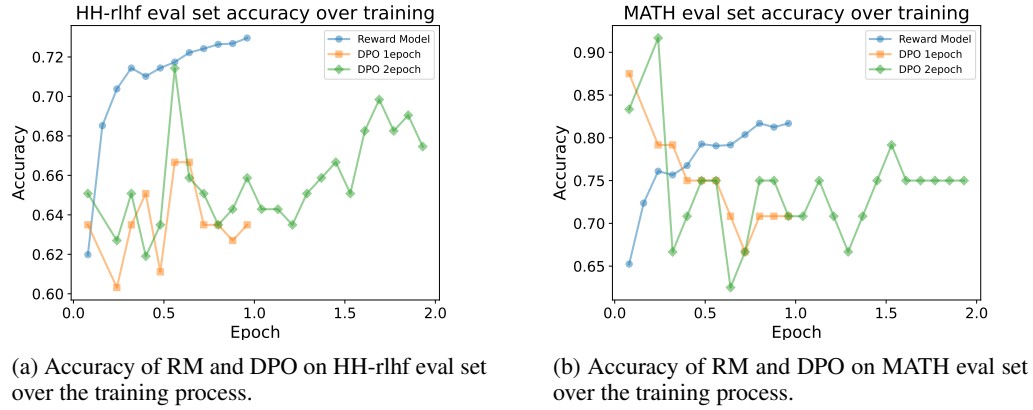

(a) Accuracy of RM and DPO on HH-rlhf eval set over the training process.

(b) Accuracy of RM and DPO on MATH eval set over the training process.

Figure 7: Comparison of DPO and RM Training. RM training demonstrates greater stability, while DPO training shows significant fluctuations.

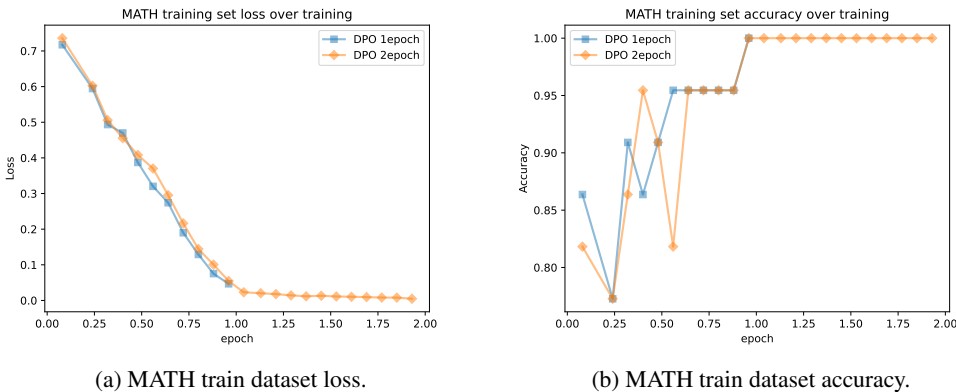

(a) MATH train dataset loss.

(b) MATH train dataset accuracy.

Figure 8: Comparison between DPO and RM training on the training set of MATH. As can be seen, in the second epoch of DPO training, the loss is very small and the accuracy of distinguishing the chosen response from the rejected response is 100%, which indicates that the model overfits the training data.

Table 7: Impact of on-policy training data on the results. $\log \pi(a^+)$ and $\log \pi(a^-)$ represent the average log probability per token for the chosen and rejected responses, respectively.

| | Baichuan2-13B | | Baichuan2-33B | |
|---|---|---|---|---|
| | $\log \pi(a^+)$ | $\log \pi(a^-)$ | $\log \pi(a^+)$ | $\log \pi(a^-)$ |
| basemodel | -0.9181 | -0.9393 | -0.3603 | -0.3634 |
| DPO in Scenario 1 | -0.9681 | -0.9982 | -0.3629 | -0.3670 |
| basemodel | -1.6776 | -0.9238 | -1.4314 | -0.3525 |
| DPO in Scenario 2 | -1.6265 | -1.2293 | -1.2734 | -0.4254 |
| basemodel | -0.9461 | -1.7110 | -0.3460 | -1.1204 |
| DPO in Scenario 3 | -0.8786 | -1.8848 | -0.3439 | -1.4333 |
| basemodel | -1.7468 | -1.7110 | -1.2838 | -1.1451 |
| DPO in Scenario 4 | -1.6617 | -1.8265 | -1.2273 | -1.2234 |

Table 8: Vanilla DPO: Baichuan2-13B and Baichuan2-33B accuracy on MATH$^*$ and SuperCLUE.

| | Baichuan2-13B | | | | Baichuan2-33B | | | |
|---|---|---|---|---|---|---|---|---|
| | MATH$^*$ | | SuperCLUE | | MATH$^*$ | | SuperCLUE | |
| | 5 points | 4&5 points | 5 points | 4&5 points | 5 points | 4&5 points | 5 points | 4&5 points |
| basemodel | 6.0% | 12.2% | 46.3% | 58.8% | 25.7% | 36.5% | 79.5% | 84.4% |
| Scenario 1 | 7.9% | 14.4% | 52.8% | 64.6% | 29.9% | 37.5% | 80.2% | 86.6% |
| Scenario 2 | 4.8% | 9.2% | 47.9% | 61.8% | 29.2% | 36.6% | 72.2% | 79.0% |
| Scenario 3 | 4.3% | 12.8% | 41.2% | 57.0% | 28.2% | 37.3% | 74.8% | 84.9% |
| Scenario 4 | 3.2% | 9.3% | 39.5% | 52.9% | 28.6% | 38.1% | 80.2% | 85.8% |

reasoning and instruction following, is limited in scope and may not generalize across all LLM applications. Additionally, the regularization techniques proposed, while effective in our controlled settings, require further validation in larger-scale models and more diverse datasets. Lastly, although we contrasted DPO with RM-based alignment, our study does not exhaustively address other potential reward-free methods, leaving open questions for future exploration.

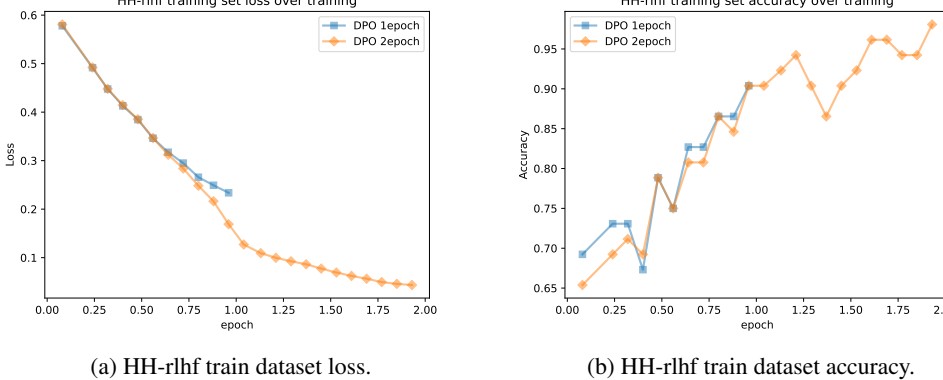

(a) HH-rlhf train dataset loss.  (b) HH-rlhf train dataset accuracy.

Figure 9: Comparison between DPO and RM training on the test set of HH-rlhf.

Table 9: DPO and its variants/regularized version performance on mathematical reasoning. In Flex-DPO, $\beta^+ = 0.1$, $\beta^- = 0.08$.

|  | MATH* | | SuperCLUE | |
|---|---|---|---|---|
|  | 5 points | 4&5 points | 5 points | 4&5 points |
| basemodel | 25.7% | 36.5% | 79.5% | 84.4% |
| DPO | 29.9% | 37.5% | 80.2% | 86.6% |
| Flex-DPO | **30.1%** | 38.0% | **81.2%** | **86.7%** |
| IPO | 30.0% | 37.7% | 80.5% | 85.9% |
| SLiC | 29.3% | **38.7%** | 79.7% | 84.5% |

