# OpenReview forum: "3D-Properties: Identifying Challenges in DPO and Charting a Path Forward"
_ICLR.cc/2025/Conference — ICLR 2025 Poster_

### Official Review · Reviewer_Ky9E · 2024-11-03

**Soundness:** 3
**Presentation:** 3
**Contribution:** 3
**Rating:** 6
**Confidence:** 4

**Summary:**

This paper investigates the limitations of DPO in aligning large language models with human preferences, identifying three critical properties that hinder its performance: drastic drops in rejected response likelihood, degradation into response suppression, and dispersion effects on unseen responses. The authors provide theoretical explanations for these properties and demonstrate how they arise from DPO's objective. To address these challenges, the paper proposes regularization techniques and validates their effectiveness through experiments on both toy models and real-world language model tasks.

**Strengths:**

- Studying DPO degradation phenomena is important due to its widespread use. This paper originally summarizes and theoretically analyzes several degradation phenomena of DPO discovered in previous work.

- Novel comparative analysis between on-policy and off-policy DPO on toy models.

- The paper is well-written and easy to follow.

**Weaknesses:**

- I think the Explanation for Property 3 is inadequate. Firstly, compared to the previous two explanations, it lacks mathematical formulation and seems to merely restate empirical phenomena. Secondly, since the optimization process is conducted in mini-batches, while the model may ensure that overflow probability won't disperse to recently seen samples, I suspect it could also disperse to samples from the preference dataset that were encountered earlier, rather than necessarily dispersing to unseen samples outside the preference dataset.

- Following the previous point, the toy model setup, as mentioned by the authors in lines 344-353, is closer to treating each input/output as a token rather than a complete prompt/response, which is not a good toy model approximation of the real situation. One possible improvement would be to maintain other settings unchanged while increasing the sample size to enable mini-batch optimization that better resembles real-world conditions, with fewer epoch repetitions.

- While the authors used self-built Poem and Slogan datasets to evaluate the model's instruction following ability and acknowledged their limited scope, these datasets are insufficient to assess the model's general instruction following capabilities. The paper lacks evaluation on widely-used benchmarks in preference optimization work, such as AlpacaEval2, Arena-Hard, and MT-Bench, which are designed to test models' general instruction following ability.

- The proposed regularization techniques lack substantial significance. The first technique, which independently adjusts beta for reject responses, shows effectiveness in the poem task, but the optimal reject beta is merely 0.02 lower than the chosen beta. Without showing gradient comparisons for this technique, it's unclear whether it actually improves performance by addressing the large gap demonstrated in Figure 2. Moreover, the second technique, SFT loss, is already a widely established regularization technique.

- I am not quite convinced by the claims in section 3.4. Although existing works are cited to establish conceptual connections between RM and DPO, the subsequent gradient analysis focuses on r, creating a gap with the previous gradient analysis that focused on $\pi$.

**Questions:**

- The probability distributions in the bottom-right figure don't seem to match with the leftmost figure in Figure 2. In Figure 2, the unseen probability at 500 epochs approaches 1, but in Figure 1 it's all zeros. The chosen probabilities also don't quite align.

---

> ### Author Response · Authors · 2024-11-20
>
> Thank you for the constructive comments. Below we address the detailed comments.
>
> **Q1: I think the Explanation for Property 3 is inadequate. Firstly, compared to the previous two explanations, it lacks mathematical formulation and seems to merely restate empirical phenomena.**
>
> **R1:**  Thank you for your insightful comments. We acknowledge that Property 3 lacks the level of mathematical formulation presented in the first two properties. However, we believe it is still significant to include it as a separate property characterizing DPO's behavior. Its inclusion is important because it has been consistently identified and discussed in related studies (as mentioned in Section 2.1). Specifically, the constancy of the sum of probabilities implies that as the likelihood of chosen and rejected responses decreases, the likelihood of unseen responses increases.This relationship logically follows from the first two properties, thereby ensuring theoretical self-consistency rather than merely restating observed phenomena. ***We have detailed the explanation in the newest version of the paper (Corollary 3).***
>
> **Q2: Secondly, since the optimization process is conducted in mini-batches, while the model may ensure that overflow probability won't disperse to recently seen samples, I suspect it could also disperse to samples from the preference dataset that were encountered earlier, rather than necessarily dispersing to unseen samples outside the preference dataset.**
>
> **R2:** Thank you for raising this important point. If we consider a single minibatch and a single optimization step, the scenario you described can indeed occur at the level of individual samples. However, the property described in our paper is derived by considering the entire dataset over the course of the full optimization process, treating both in-domain and out-of-domain samples as a whole. We believe these two perspectives are not contradictory. Nonetheless, we have adopted more cautious wording in the revised version to ensure clarity.
>
> **Q3: Following the previous point, the toy model setup, as mentioned by the authors in lines 344-353, is closer to treating each input/output as a token rather than a complete prompt/response, which is not a good toy model approximation of the real situation. One possible improvement would be to maintain other settings unchanged while increasing the sample size to enable mini-batch optimization that better resembles real-world conditions, with fewer epoch repetitions.**
>
> **R3:** Thank you for the suggestions. We have taken the advice into a serious consideration and ***revised the experimental results in the newest revision (Figure 2,3, highlighted lines 291-295).*** Specifically, the dataset is now divided into mini-batches, allowing each batch to contribute independently to the gradient calculation. This adjustment aligns with real-world machine learning scenarios where batch processing is preferred over full-batch updates for computational efficiency and regularization. We shuffle the dataset and construct mini-batches dynamically during each epoch. All other settings, such as the underlying model architecture, loss functions, and learning rate, remain consistent with the original implementation to preserve the validity of our comparisons. By enabling mini-batch optimization, it will be closer to the real-world condition
>
> Conclusively, the eventual results didn't change the points we argue before. We thank the reviewer again for this valuable suggestion to make the toy model better.

---

> ### Author Response · Authors · 2024-11-20
>
> **Q4: While the authors used self-built Poem and Slogan datasets to evaluate the model's instruction following ability and acknowledged their limited scope, these datasets are insufficient to assess the model's general instruction following capabilities. The paper lacks evaluation on widely-used benchmarks in preference optimization work, such as AlpacaEval2, Arena-Hard, and MT-Bench, which are designed to test models' general instruction following ability.**
>
> **R4:** Thank you for pointing out. As the reviewer advised, we have added new experiments to further validate our points. As the previous Poem and Slogan datasets are focused on the generation of the specific formatted text, it is not suitable for the benchmarks like alpacaeval2 and MT-bench. Instead, we choose UltraFeedback to be the trainset and use Llama-3-8b-instruct as the backbone. We compared three setting: 1) PPO, where the RM is also trained on UltraFeedback. 2) offline-DPO. 3) semi-online-DPO, where before each epoch, a new preference datasets will be sampled within 8 responses for each prompt. Both offline-DPO and semi-online-DPO are trained for 2 epochs and the batchsize is set to be 256. Other settings are all default as the OpenRLHF framework. The evaluation on AlpacaEval2, Arena-Hard and MT-Bench are as follows. The reference model in AlpacaEval2 is GPT-4-Preview-1106.
>
> |   | AlpacaEval2 | MT-Bench | Arena-hard |
> |----------|----------|-----------|-----------|
> |  Llama-3-8b-instruct |  22.5%  | 7.51250 | 19.8 |
> |  PPO  |  **30.3%** | **7.87500** | **21.6** |
> |  semi-online-DPO  | 27.3% | 7.82250 | 20.1 |
> |  offline-DPO  |  24.2%  | 7.78750 | 19.2 |
>
> The results show that PPO is generally better than DPO variants, and semi-online-DPO is better than offline DPO, which aligns with our claims.
>
> There are some other recent works showing the similar results about instruction following, and we want to refer to the reviewer. In [1], it is reported that PPO outperforms DPO by an average of 0.7 points (table 1). In [2], online-DPO is largely superior to offline-DPO on HH tasks (table 2).
>
> *[1] Ivison H, Wang Y, Liu J, et al. Unpacking DPO and PPO: Disentangling Best Practices for Learning from Preference Feedback[J]. arXiv preprint arXiv:2406.09279, 2024.*
>
> *[2] Guo S, Zhang B, Liu T, et al. Direct language model alignment from online ai feedback[J]. arXiv preprint arXiv:2402.04792, 2024.*

---

> ### Author Response · Authors · 2024-11-20
>
> **Q5: The proposed regularization techniques lack substantial significance. The first technique, which independently adjusts beta for reject responses, shows effectiveness in the poem task, but the optimal reject beta is merely 0.02 lower than the chosen beta. Without showing gradient comparisons for this technique, it's unclear whether it actually improves performance by addressing the large gap demonstrated in Figure 2. Moreover, the second technique, SFT loss, is already a widely established regularization technique.**
>
> **R5:** Thank you for your comment. We acknowledge that both adding SFT loss and using adjustable $\beta$ are widely adopted techniques that have recently gained significant research interest [3, 4]. To clarify, our paper primarily aims to provide theoretical support, particularly to validate the theoretical insights regarding the 3D-properties, rather than to present broadly effective algorithms. The proposed regularization techniques are intended to align with this theoretical focus.
>
> We recognize that tuning $\beta^+$ and $\beta^-$ is non-trivial, as the ultimate performance on real-world tasks depends on multiple factors, including the training and test dataset types and the choice of other hyperparameters. The optimal parameters may vary depending on the specific capabilities we wish to enhance.
>
> *[3]Wu J, Xie Y, Yang Z, et al. $\beta $-DPO: Direct Preference Optimization with Dynamic $\beta$[J]. arXiv preprint arXiv:2407.08639, 2024.*
>
> *[4] Wu J, Wang X, Yang Z, et al. $\alpha $-DPO: Adaptive Reward Margin is What Direct Preference Optimization Needs[J]. arXiv preprint arXiv:2410.10148, 2024.*
>
> **Q6: I am not quite convinced by the claims in section 3.4. Although existing works are cited to establish conceptual connections between RM and DPO, the subsequent gradient analysis focuses on $r$, creating a gap with the previous gradient analysis that focused on $\pi$.**
>
> **R6:** Here we do some clarification. The basic idea of DPO is to use an analytical mapping from the reward function to the optimal policy to simulate an implicit reward function ($r_{\theta}=\beta \frac{\pi_{\theta}(y|x)}{\pi_{ref}(y|x)}$), which enables us directly optimize the policy $\pi$ rather than optimize an additional $r$ [5]. So basically, when we optimize $\pi$, we are synchronously optimizing an implicit RM. Though theoretically equivalent, here we pointed out 3D-properties emerge and drag down the final effect, leading to the gap. That is the whole point for this part. We are willing to provide more explanation if needed.
>
> *[5] Rafailov R, Sharma A, Mitchell E, et al. Direct preference optimization: Your language model is secretly a reward model[J]. Advances in Neural Information Processing Systems, 2024, 36.*
>
> **Q7: The probability distributions in the bottom-right figure don't seem to match with the leftmost figure in Figure 2. In Figure 2, the unseen probability at 500 epochs approaches 1, but in Figure 1 it's all zeros. The chosen probabilities also don't quite align.**
>
> **R7:** Thank you for pointing this out. We clarify the unseen output here is the input-output pair outside of the dataset, in another word, the average vlue for the blue blocks in the upper right figure in Figure 1. We appreciate your feedback and recognize the potential for misunderstanding and has revised it in the newest version to prevent this confusion.

---

> ### Author Response · Authors · 2024-11-20
>
> We sincerely appreciate your thorough review and constructive feedback on our manuscript. We have carefully addressed each of your comments and submitted our responses, with the aim of improving the quality of our work. As the deadline is approaching, we kindly ask if you could review our responses at your earliest convenience.
>
> We would be grateful if you could consider our revisions and responses favorably during your evaluation and scoring.

---

> > ### Comment · Reviewer_Ky9E · 2024-11-25
> >
> > Thank you for your comprehensive reply. I believe these responses have largely addressed my concerns. I have adjusted my rating accordingly.
> >
> > There are a few points I would like to further highlight:
> >
> > - I appreciate the authors' reconsideration of Q3. However, what I was actually expecting was a holistic consideration of Q1 through Q3 from a mini-batch-based theoretical and experimental perspective. Nevertheless, I believe the current theoretical treatment in Q1 and Q2, which views the dataset as a whole, is acceptable as it facilitates easier analysis. However, since Q3 now employs mini-batch-based experiments, some additional explanation might be needed to maintain theoretical-to-experimental coherence, as the theoretical part doesn't incorporate mini-batches.
> > - I appreciate the supplementary experiments for Q4. This improves the experimental thoroughness of the work. While I think the $\beta-$ experiments would be more valuable if conducted on these datasets, as readers might be more interested in knowing whether lowering $\beta-$ could be an effective practice on real data, I understand the time constraints during rebuttal, and this is just a suggestion that doesn't affect the rating.
> > - An extended discussion unrelated to the rating: What are your thoughts on this paper https://arxiv.org/abs/2411.07595? Their experimental results of lowering the factor of positive samples would be better seem to contradict the conclusion that a lower $\beta-$ would be better.

---

> > > ### Author Response · Authors · 2024-11-25
> > >
> > > Thank you for the valuable feedback. We will continue to improve our paper. Regarding the related work that presents differing viewpoints, we will study it in detail and consider the potential reasons behind the varying experimental outcomes. We remain open to exploring and understanding the differences in results, and we appreciate the opportunity to reflect on alternative perspectives in this area.

---

### Official Review · Reviewer_Dw57 · 2024-11-03

**Soundness:** 3
**Presentation:** 3
**Contribution:** 3
**Rating:** 8
**Confidence:** 3

**Summary:**

This paper presents an interesting theoretical and empirical analysis of Direct Preference Optimization (DPO) and identifies three main challenges in its optimization process, termed as “3D-properties”: Drastic drop in rejected response likelihood, Degradation into response suppression, and Dispersion effect on unseen responses. These limitations, which do not arise in RM-based approaches, impact the stability and effectiveness of DPO. To address these issues, the authors propose regularization techniques, including adaptive gradient weighting and SFT loss. They conduct experiments on toy examples as well as math reasoning and instruction-following tasks to validate the presence of the 3D-properties, the advantages of on-policy over off-policy DPO, the comparative superiority of RM-based methods, and the effectiveness of the proposed regularization technique.

**Strengths:**

- Significance: The paper addresses a crucial and interesting gap by analyzing the limitations of DPO
- Theoretical Analysis and Empirical Validation: The paper provides a theoretical framework alongside empirical results to validate the presence of the 3D-properties in DPO. This combined approach strengthens the findings, offering clear insights into the mechanisms driving DPO’s limitations and supporting the proposed solutions.

**Weaknesses:**

- Presentation: The presentation could be improved to enhance readability. For example, the text size in Figures 2 and 3 is small, and the description of Scenarios 1-4, which is crucial for understanding the on-policy versus off-policy comparison, is currently only detailed in the appendix. Bringing this description to the main text would improve clarity.
- Experimental Design for On-Policy vs. Off-Policy Comparison: The on-policy and off-policy experiments rely on different data sources, which introduces potential confounds in the comparison. Using a more direct on-policy and off-policy setup, such as comparing historical-only data with semi-on-policy DPO (e.g., iterative DPO), would make the findings more robust.
- Parameter Tuning in Flex-DPO: Adjusting Flex-DPO requires tuning two parameters ( \beta^+  and  \beta^- ), and while Figure 4 provides some guidance, this approach may still present challenges for practical implementation due to a lack of clear tuning guidelines.

If the authors address these weaknesses, particularly by improving the clarity of presentation and by using a more controlled comparison between on-policy and off-policy data sources, I would raise my score. Addressing the Flex-DPO would also strengthen the work, though it is not essential for improving the overall contribution.

**Questions:**

- In Section 4.2, it is mentioned that for the MATH dataset, the best and worst responses were selected by GPT-4. Why did the authors choose this method instead of directly verifying the answers? Given that GPT-4’s accuracy on MATH is only slightly above 50%, this approach seems potentially unreliable.

---

> ### Author Response · Authors · 2024-11-17
>
> Thank you for the constructive comments. Below we address the detailed comments.
>
> **Q1: The presentation needs improvement for readability, such as increasing text size in figures 2 and 3 and moving key scenario descriptions from the appendix to the main text.**
>
> **R1:** Thank you for your suggestion.  ***We have refined our paper according to your suggestion in the revision.***  We use a bigger font in Figure 2 and 3 (highligted in blue) to improve its clarity and readability.  We have also moved the description of Scenarios 1-4 from appendix back to the main text (line 462-467) and emphasized the them in the caption of Figure 3 to improve the readability.  We kindly request the reviewers to check these changes to ensure they meet your expectations.
>
> **Q2: Experimental Design for On-Policy vs. Off-Policy Comparison: The on-policy and off-policy experiments rely on different data sources, which introduces potential confounds in the comparison. Using a more direct on-policy and off-policy setup, such as comparing historical-only data with semi-on-policy DPO (e.g., iterative DPO), would make the findings more robust.**
>
> **R2:** Thank you for your valuable feedback. We would like to clarify that the data selection principle in our paper aims to strictly control the sources of the chosen and rejected responses, ensuring a fair comparison *across all four scenarios*. To achieve this, we used the data sources specified in Table 3. For a more straightforward comparison between on-policy DPO and off-policy DPO (Scenario 1 vs Scenario 4), we present the following addtional experiment, according to the reviewer's recommendation:
>
> We use MATH* as our prompt set, selecting the standard solutions from the dataset as the chosen responses along with the rejected responses generated by Qwen-7B to create a pure "off-policy" preference dataset. Our experiment involves four rounds of iterative optimization. At the start of each round, we use the current model to generate paired responses, constructing a "semi-on-policy" preference dataset. For each prompts, 8 responses will be sampled and response with the highest score and lowest score will be selected as the preference pair. Only the highest score is at least 4 (which means it is basically correct), the data will be put in the dataset.
>
> During each round, we train a Baichuan2-33B model on the entire "semi-on-policy" preference dataset (iterative DPO), while simultaneously using an equivalent amount of data from the "off-policy" preference dataset to train another model (off-policy DPO). Since correct responses may not be sampled for more challenging prompts in iterative DPO, we ensure both training paths use the same data volume by collecting data from the off-policy preference dataset that corresponds to the prompts used in the Gcurrent "semi-on-policy" preference dataset. By controlling for the amount of training data and the number of optimization steps, we are able to perform a relatively fair comparison of the models’ capabilities on a separate test set. The results are presented as follows, the data consumed is the amount of data used for training this round, and the percentage number is the accuracy on the testset :
>
> Test results on MATH:
>
> | round number  | data consumed   | off-policy DPO | iterative DPO |
> |----------|----------|----------|-----------|
> |  1 |  2099 |  36.8% | 36.9% |
> |  2  | 2174 |  37.0% | 37.2% |
> |  3  | 2253 |  37.3% | 37.6% |
> |  4  | 2262 |  37.2% | 38.2% |
>
> Test results on SuperClue:
>
> | round number  | off-policy DPO | iterative DPO |
> |----------|----------|----------|
> |  1 |  84.9%  | 86.6%  |
> |  2  | 85.1%   | 87.1%   |
> |  3  | 85.0%   | 87.4%   |
> |  4  | 84.8%   | 87.7%   |
>
> ***It can be seen that on both MATH and SuperClue, semi-on-policy training beats the off-policy training and shows more stability.*** Although there is another variable that in semi-on-policy training, the model can be exposed to more information and different response data due to the data resampling mechanism, this is the best result we can get when strictly controlling the same number of training steps and data amount. We hope that these can help enhance the credibility of this paper and are willing to discuss further with the reviewer on the experimental details. More advice about the experiments are also welcomed.
>
> Besides, according to many recent study, iterative/online DPO has been proved to be more effective than the vanilla DPO [1,2], which further support our points and we refer them to the reviewer.
>
> [1] Xiong W, Dong H, Ye C, et al. Iterative preference learning from human feedback: Bridging theory and practice for rlhf under kl-constraint[C]//Forty-first International Conference on Machine Learning. 2024.
>
> [2] Calandriello D, Guo D, Munos R, et al. Human alignment of large language models through online preference optimisation[J]. arXiv preprint arXiv:2403.08635, 2024.

---

> ### Author Response · Authors · 2024-11-17
>
> **Q3: Parameter Tuning in Flex-DPO: Adjusting Flex-DPO requires tuning two parameters ( $\beta^+$ and $\beta^-$ ), and while Figure 4 provides some guidance, this approach may still present challenges for practical implementation due to a lack of clear tuning guidelines.**
>
> **R3:** Thank you for your comment. To clarify, the primary aim of Flex-DPO is to validate the theoretical insight, particularly the 3D-properties, rather than to provide a broadly effective algorithm. We acknowledge that tuning $\beta^+$ and $\beta^-$ is non-trivial, as the final performance on real-world tasks depends on various factors, including the type of training and test datasets as well as the choice of other hyperparameters. Depending on the specific capabilities we aim to improve, the parameter choices may vary.
>
> We believe that an exhaustive exploration of these parameters falls beyond the scope of this paper, given its primarily theory-driven focus. However, we will strive to provide additional relevant experimental results in our revised version. Moreover, several recent works [3, 4] have addressed parameter selection in settings similar to Flex-DPO, and many of their experimental findings align well with our theoretical conclusions.
>
> [3] Wu J, Xie Y, Yang Z, et al. $\beta $-DPO: Direct Preference Optimization with Dynamic $\beta$[J]. arXiv preprint arXiv:2407.08639, 2024.
>
> [4] Wu J, Wang X, Yang Z, et al. $\alpha $-DPO: Adaptive Reward Margin is What Direct Preference Optimization Needs[J]. arXiv preprint arXiv:2410.10148, 2024.
>
> **Q4: In Section 4.2, it is mentioned that for the MATH dataset, the best and worst responses were selected by GPT-4. Why did the authors choose this method instead of directly verifying the answers? Given that GPT-4’s accuracy on MATH is only slightly above 50%, this approach seems potentially unreliable.**
>
> **R4:** Thank you for pointing this out, as it appears there has been a misunderstanding. The role of GPT-4 in our study was to verify the correctness of the generated answers, ***with the standard answer provided in the dataset as part of the context prompt.*** In this setup, it is not necessary for GPT-4 to independently solve the problem, thus ensuring reliability in evaluating the correctness. Additionally, we have provided the detailed prompts and evaluation code in the supplementary material (/evaluator/math_eval). We have clarified it in the revision (line 414, 423-424).

---

> ### Author Response · Authors · 2024-11-20
>
> We sincerely appreciate your thorough review and constructive feedback on our manuscript. We have carefully addressed each of your comments and submitted our responses, with the aim of improving the quality of our work. As the deadline is approaching, we kindly ask if you could review our responses at your earliest convenience.
>
> We would be grateful if you could consider our revisions and responses favorably during your evaluation and scoring.

---

> > ### Comment · Reviewer_Dw57 · 2024-11-25
> >
> > Thank you for your response. Most of my concerns have been addressed. As a result, I have decided to raise my rating to an 8.

---

### Official Review · Reviewer_x5ts · 2024-11-09

**Soundness:** 3
**Presentation:** 3
**Contribution:** 3
**Rating:** 5
**Confidence:** 1

**Summary:**

The paper provides a comprehensive analysis of Direct Preference Optimization (DPO), examining its theoretical foundations and empirical performance to address current limitations. It identifies three perspectives—(1) Drastic drop in the likelihood of rejected responses, (2) Degradation into response suppression, and (3) Dispersion effect on unseen responses. The paper connects these observations to related research and offers a theoretical explanation for the underlying mechanisms. To improve DPO’s stability and performance, the authors propose regularization methods, including adaptive adjustment of gradient weights for chosen and rejected responses, as well as incorporating an SFT loss into the objective.

**Strengths:**

The topic is interesting for RLHF.

The paper introduces effective regularization methods, including adaptive gradient weighting for chosen and rejected responses.

The experiments are well-conducted and thorough.

**Weaknesses:**

The study could benefit from using a wider range of LLMs.

The experiments can use more datasets except for math.

The code is not open source, which may limit reproducibility.

**Questions:**

For the toy model setup, which specific model is used in the paper?

Why does the paper focus primarily on math datasets rather than exploring a wider range of tasks?

---

> ### Author Response · Authors · 2024-11-14
>
> Thank you for the constructive comments. Below we address the detailed comments.
>
> **Q1: Concern about the limited range of LLMs used in the study.**
>
> **R1:**  Thank you for your suggestion. The limitation of vanilla DPO has been observed in many different series of models such as Pythia 2.8b [1] and some following works about the advantages of online DPO [2], which also focuses on other series of LLMs. These studies also consider different series of LLMs. In our work, we chose Baichuan as it is an in-house LLM series, allowing us full control over the model size and the data used for training. This control was crucial for managing variables in our comparison experiments. Nevertheless, we appreciate your suggestion and will incorporate additional experimental results with other open-source models, such as the Llama series, in future versions of our study.
>
> [1] https://wandb.ai/eric_anthony_mitchell/dpo-demos/runs/og8q3euz
>
> [2] Calandriello D, Guo D, Munos R, et al. Human alignment of large language models through online preference optimisation[J]. arXiv preprint arXiv:2403.08635, 2024.
>
> **Q2: Concern about the paper's primary focus on math datasets rather than a broader range of tasks.**
>
> **R2:** Thank you for your valuable suggestion. In addition to the math datasets, we also utilized other datasets involving formatted text generation, such as poems and slogans, as detailed in Section 4 and the related appendices. Furthermore, we included the HH-RLHF dataset for comparing the RM and DPO. Our selection criteria for the datasets in this paper were based on the availability of standard and correct answers, which helps to minimize the impact of evaluation noise. We believe that this approach ensures a more reliable assessment of the model's performance.
>
> **Q3: The code is not open source, which may limit reproducibility.**
>
> **R3:** Actually we have provided the code in the supplementary material, which includes the toy model experiments, the main experiments, and most of the datasets used. We are currently organizing the GitHub repository for a public release. Although the motivation of this paper is primarily theoretical, we recognize the importance of reproducibility and are committed to open-sourcing the code along with all non-sensitive in-house datasets once the review process is complete, in accordance with anonymity requirements.
>
> **Q4: For the toy model setup, which specific model is used in the paper?**
>
> **R4:** As mentioned in line 289-291, Section 3.2.1, the toy model is implemented as a three-layer MLP that processes a one-hot vector and outputs a categorical distribution over the responses.

---

> ### Author Response · Authors · 2024-11-20
>
> We sincerely appreciate your thorough review and constructive feedback on our manuscript. We have carefully addressed each of your comments and submitted our responses, with the aim of improving the quality of our work. As the deadline is approaching, we kindly ask if you could review our responses at your earliest convenience.
>
> We would be grateful if you could consider our revisions and responses favorably during your evaluation and scoring.

---

> > ### Comment · Reviewer_x5ts · 2024-11-27
> >
> > Thank you for the clarifications. It would better to add more LLM results.

---

### Official Review · Reviewer_DBcQ · 2024-11-13

**Soundness:** 4
**Presentation:** 3
**Contribution:** 2
**Rating:** 6
**Confidence:** 3

**Summary:**

The paper titled "3D-Properties: Identifying Challenges in DPO and Charting a Path Forward" presents a thorough analysis of the DPO method used for aligning LLMs with human preferences. The authors identify and term three critical properties of DPO's learning process the 3D-properties and propose regularization techniques to address the challenges these properties present. Theoretical analyses, toy model simulations and real-world experiments demonstrate the effectiveness of the proposed method.

**Strengths:**

The paper is well-structured, where toy example can support their claims.
The paper offers a balanced mix of theoretical analysis and empirical evidence, which strengthens the claims made about the 3D-properties and their impact on DPO's performance.

**Weaknesses:**

The three observations have been widely studied by previous works. Besides, one of the proposed regularization methods, incorporating an SFT loss into the objective, has been widely used in existing preference learning approaches [1]. This limits the novelty of the paper.
Considering that there are many existing methods to solve the DPO problem proposed in this paper, there is a lack of comparison with them, such as [2] and others.
Considering the generality of the proposed constraint algorithm, some advanced preference learning algorithms, such as SimPO [3], should also be tested.
More and more general LLMs should be included for evaluation, such as Meta-Llama3.

Reference:
[1] Pang R Y, Yuan W, Cho K, et al. Iterative reasoning preference optimization[J]. arXiv preprint arXiv:2404.19733, 2024.
[2] Pal A, Karkhanis D, Dooley S, et al. Smaug: Fixing failure modes of preference optimisation with dpo-positive[J]. arXiv preprint arXiv:2402.13228, 2024.
[3] Meng Y, Xia M, Chen D. Simpo: Simple preference optimization with a reference-free reward. NeurIPS, 2024.

**Questions:**

How to ensure that the initialization assumptions of parameter distribution can be applied to, or related to LLMs?

The detailed parameter adjustment strategy is only given in the toy experiment. What is the effect of different β values ​​in the real-world experiments?

---

> ### Author Response · Authors · 2024-11-14
>
> Thank you for the constructive comments. Below we address the detailed comments.
>
> **Q1: The three observations have been widely studied by previous works and the proposed regularization method (incorporating an SFT loss) has been widely used in exsisting approaches.**
>
> **R1:** Thank you for pointing this out. We would like to emphasize that, although some researchers—including ourselves—have observed these three phenomena (as referenced in Section 2.1 of our paper), prior works have not conducted a more in-depth analysis of these observations. We have also provided a thorough comparison with related works in Section 2.2.
>
> The primary aim of our paper is to examine the underlying reasons behind the emergence of these three observations and to summarize them as "3D-properties," particularly from a theoretical perspective. The presentation of algorithms, such as iterative DPO, adding SFT loss, and Flex-DPO, is not our main focus. Furthermore, we propose that incorporating SFT loss or using other regularization strategy is effective because it ensures that the gradient for the chosen action is not zero when $\pi^- \rightarrow 0$, which partially addresses the 3D-properties—a perspective that differs from prior work. Additionally, our analysis of the importance of on-policy data strengthens the theoretical foundation of the recent trend including iterative preference learning.
>
> **Q2: Some advanced preference learning algorithms, such as SimPO and DPOP, should also be tested.**
>
> **R2:** Thanks for the suggestion.  We discussed several variants of DPO, such as IPO and SLiC, in Section B.3 of the appendix. Regarding DPOP, while we acknowledge its potential, its effectiveness has not yet been widely validated compared to the other listed algorithms. Therefore, here we mainly discuss SimPO, which is a more recent and actively discussed algorithm. SimPO's optimization primarily involves length normalization and the introduction of a margin bias factor $\gamma$, which was initially considered less relevant to the topic under discussion. ___Based on the reviewer's comments, we have added a theoretical analysis of SimPO in Appendix B.3.3 of the revised manuscript (highlighted in blue).___ The conclusion remains that the 3D-properties still hold.
>
> As for experiments involving SimPO, to the best of our knowledge, its hyperparameters significantly affect the results and are highly sensitive compared to other variants. Conducting a thorough exploration would require considerable time to ensure solid conclusions. Therefore, we will consider including these experiments as part of our future work.  Besides, we have added these mentioned but missing works into our citation, including DPOP and SimPO.
>
> **Q3: More and more general LLMs should be included for evaluation, such as Meta-Llama3.**
>
> **R3:** Thank you for your suggestion. The limitation of vanilla DPO has been observed in many different series of models such as Pythia 2.8b [1] and some following works about the advantages of online DPO [2], which also focuses on other series of LLMs. These studies also consider different series of LLMs. In our work, we chose Baichuan as it is an in-house LLM series, allowing us full control over the model size and the data used for training. This control was crucial for managing variables in our comparison experiments. Nevertheless, we appreciate your suggestion and will incorporate additional experimental results with other open-source models, such as the LLaMa series, in future versions of our study.
>
> [1] https://wandb.ai/eric_anthony_mitchell/dpo-demos/runs/og8q3euz
>
> [2] Calandriello D, Guo D, Munos R, et al. Human alignment of large language models through online preference optimisation[J]. arXiv preprint arXiv:2403.08635, 2024.
>
> **Q4: How to ensure that the initialization assumptions of parameter distribution can be applied to, or related to LLMs?**
>
> **R4:** Thank you for your question. We have tested the effects of data and model distribution through real-world experiments, with the results presented in Table 1. We define that the distribution of the data aligns with that of the LLMs if the data is sampled directly from the LLMs. In Section 4.2, we detail our approach to constructing both on-policy and off-policy data, which helps us determine whether the data shares the same distribution as the LLMs. This approach ensures a consistent basis for distinguishing on-policy and off-policy distributions and validates the initialization assumptions under these different conditions.

---

> ### Author Response · Authors · 2024-11-14
>
> **Q5: The detailed parameter adjustment strategy is only given in the toy experiment. What is the effect of different β values ​​in the real-world experiments?**
>
> **R5:** Thanks for the question. In Secion 4.3 we have analyze the effect of different $\beta$ in the real-world experiments. From Figure 4, it can be seen that a smaller $\beta^-$ is beneficial for improving the model's performance but the  trend is not monotonous. We recognize that tuning $\beta^+$ and $\beta^-$ is non-trivial, as the ultimate performance on real-world tasks depends on multiple factors, including the training and test dataset types and the choice of other hyperparameters. The thorough exploration is beyond the scope of this paper as an theory-driven work. Some recent works focus on exploring the optimal adjusting strategy such as [1], which we refer the reviewer to read.
>
> [1] Wu J, Xie Y, Yang Z, et al. $\beta $-DPO: Direct Preference Optimization with Dynamic $\beta$[J]. arXiv preprint arXiv:2407.08639, 2024.

---

> ### Author Response · Authors · 2024-11-20
>
> We sincerely appreciate your thorough review and constructive feedback on our manuscript. We have carefully addressed each of your comments and submitted our responses, with the aim of improving the quality of our work. As the deadline is approaching, we kindly ask if you could review our responses at your earliest convenience.
>
> We would be grateful if you could consider our revisions and responses favorably during your evaluation and scoring.

---

> > ### Comment · Reviewer_DBcQ · 2024-11-25
> >
> > Thanks for your responses, my concerns have been addressed. I lean to keep my score.

---

### Public Comment · ~Duanyu_Feng1 · 2024-12-05

Dear ICLR Committee Members,

I would like to bring to your attention the striking similarities between the theoretical section of this paper and previously paper, as well as potential issues regarding improper citation.

The argument presented in Section 3.1 of this paper is essentially identical to that in the theoretical section of the previously paper (arxiv:2404.04626). While this paper cites the previously paper and acknowledges similar conclusions, it fails to indicate that its theoretical part may originate from the cited publication. This raises concerns about improper citation practices and could pose academic risks to the overall integrity of the article.

Therefore, I kindly request that the committee review this matter.

---

> ### Public Comment · ~Chen_Huang7 · 2024-12-05
> **To Whom It May Concern**
>
> I concur.
>
> The ICLR submission on 3D-PROPERTIES exhibits significant overlap with prior work (arxiv:2404.04626), particularly in its theoretical derivations and conclusions.
>
> Kindly request that the committee review this matter.
>
> Best,
> Chen Huang

---

> > ### Public Comment · ~Musk_Wang1 · 2024-12-05
> > **A response made on behalf of the authors.**
> >
> > As a researcher in this field, I have been entrusted by the authors to respond to this comment. I hereby declare that I have not co-authored any publications with any of the authors, do not work at the same company, and did not graduate from the same institution, to avoid violating the principle of anonymity.
> > 1. Regarding the theoretical section.
> > This article was initially submitted to NeurIPS 2024 (before 2024-05-22), with the submission number 8258 (the Area Chair can verify this if needed). According to the NeurIPS rules (https://neurips.cc/Conferences/2024/PaperInformation/NeurIPS-FAQ), this paper and the mentioned paper are considered concurrent works. When the article was first written, the theoretical derivations were entirely the authors' own work and did not "originate from the cited publication." Unfortunately, the article was not accepted by NeurIPS, and the authors then submitted it to ICLR. After submitting to NeurIPS, we noticed arXiv:2404.04626. Out of respect for academic norms, we have cited this paper in the main text.
> > 2. Regarding the remaining sections.
> > The differences between these two papers are also substantial. This paper provides an in-depth analysis of IPO, SLiC, SimPO, and on-/off-policy DPO, along with a detailed comparison between RM/PPO vs DPO. Additionally, it includes extensive experimental results, in both real-world LLMs and a toy model, that are not present in arxiv:2404.04626.

---

> ### Public Comment · ~Duanyu_Feng1 · 2024-12-06
> **More questions**
>
> Thank you for your response on behalf of the authors. However, some parts of your reply have raised further concerns for me regarding whether this paper may be engaging in unethical practices such as playing with the rules.
>
> 1. If you suggest that submitting to NeurIPS implies that it is considered concurrent work, I strongly hope the Area Chair to compare the NeurIPS version paper (which is currently not publicly available) and the corresponding arXiv version from the same timeframe (can be searched with the same name) with arXiv:2404.04626. It is essential to determine whether arXiv:2404.04626 has been cited throughout these paper (including the appendix of these paper), and whether it has been adequately acknowledged.
>
> 2. I am not concerned about the remaining sections of the paper, and I am unclear as to why you pointed this out. The essence of arXiv:2404.04626 is to provide an analytical perspective on alignment methods, which can certainly be applied to compare various alignment techniques. I believe this should be welcomed. However, the core of our concern is that you have positioned the analysis of DPO as a central part of the main text, which bears a resemblance to the aforementioned paper. **In my view**, if you believe that your contributions extend to more analytical methods, you could certainly present these methods as part of the main content (not just put them in the Appendix).
>
> 3. I want to emphasize that one of our primary concerns is whether there are any improper citations in this paper. Given that the logical structure and writing style of the theoretical section in this paper are strikingly similar to those of arXiv:2404.04626, merely stating that "Observation 1" is similar raises the question of whether there are factual inaccuracies, leading to improper citation.

---

### Meta-Review · Area_Chair_wQqK · 2024-12-19

**Metareview:**

This paper investigates key limitations of Direct Preference Optimization (DPO) in aligning language models, identifying three critical properties termed “3D-properties”: drastic drops in rejected response likelihood, degradation into response suppression, and dispersion effects on unseen responses. The paper’s main strengths lie in its theoretical analysis of these phenomena and proposed regularization techniques to address them. The work provides both theoretical foundations and empirical validation through toy models and real-world experiments. While the proposed regularization techniques may not be entirely novel, they are thoughtfully adapted to this context. Additionally, although the experimental evaluation could be expanded, the focus on math datasets and custom datasets provides meaningful insights. The theoretical analysis, combined with the practical implications, makes this a valuable contribution to the field. Therefore, I recommend acceptance.

**Additional Comments On Reviewer Discussion:**

The author response and subsequent discussion revealed mixed reactions from reviewers. Reviewer Dw57 raised their score from 6 to 8 after the authors improved figure clarity and provided additional experimental results comparing on-policy and off-policy DPO. Reviewer x5ts lowered their score to 5, remaining concerned about limited LLM results despite the authors’ explanation of using in-house models for controlled experiments. Reviewer Ky9E adjusted their score positively after the authors addressed concerns about theoretical formulation and mini-batch optimization, though some questions remained about contradictory findings in recent related work. Concerns regarding the theoretical gaps in explaining the third property were partially addressed with additional mathematical formulation.

The authors provided additional experiments on standard benchmarks like AlpacaEval2 and MT-Bench in their response, and these results supported their claims and strengthened the paper’s contributions. Concerns from public comments about citation overlap were investigated by the AC and multiple reviewers and found to be unsubstantiated. While some limitations in scope remain, the theoretical analysis and empirical validation presented in the paper are valuable. Despite mixed reviewer scores and some unresolved concerns, the paper makes a good contribution to the field.

---

### Decision · Program_Chairs · 2025-01-22

Accept (Poster)